# Antarctic sensitivity to oceanic melting parameterizations

Antonio Juarez-Martinez[1,2], Javier Blasco[3], Alexander Robinson[4], Marisa Montoya[1,2], and
Jorge Alvarez-Solas[1,2]

[1]Complutense University of Madrid, Madrid, Spain
[2]Geoscience Institute CSIC-UCM, Madrid, Spain
[3]Université Libre de Bruxelles, Brussels, Belgium
[4]Alfred Wegener Institute, Helmholtz Centre for Polar and Marine Research, Potsdam, Germany

**Correspondence:** Antonio Juarez-Martinez (antjua01@ucm.es)

**Abstract.** The Antarctic Ice Sheet (AIS) has experienced accelerated loss of ice over the last decades and could become the main contributor to sea-level rise in the coming centuries. However, the associated uncertainty is very large. The main sources of this uncertainty lie in the future scenarios, the climatic forcing and, most notably, the structural uncertainty due to our lack of understanding of ice-ocean interaction processes, in particular, the representation of sub-shelf basal melt. In this study, we use a higher-order ice-sheet model to investigate the impact of these three sources of uncertainty in the contribution of the AIS to sea level in the coming centuries in the context of the Ice Sheet Model Intercomparison Project for CMIP6 (ISMIP6) but extending the projections until 2500. We test the sensitivity of the model to basal melting parameters using several forcings and scenarios simulated in the CMIP5 and CMIP6 ensembles. Results show a strong dependency on the values of the parameter that controls the heat exchange velocity between ice and ocean and also the forcing and scenario. Higher values of the heat exchange parameter lead to higher sea-level rise, with the contribution depending on the forcing-scenario configuration and reaching in some cases more than 3 meters in sea level equivalent by the end of 2500. Idealized simulations considering the individual effects of the atmospheric and the oceanic forcing have been performed, demonstrating that the oceanic forcing plays a dominant role over the western sector of the AIS while the atmospheric forcing is more important for the eastern sector and the interior.

## 1 Introduction

Antarctica hosts today's largest ice sheet on Earth, the Antarctic Ice Sheet (AIS), with a total volume close to 27 million km$^3$, corresponding to a sea level equivalent (SLE) of about 58 m (Morlighem et al., 2020). Ice flows from the ice-sheet interior within ice streams towards the ice-sheet margin. As it spreads outwards, it thins and eventually starts floating, forming ice shelves. The AIS is fringed by more than 300 of them, representing nearby 2.5% of its volume of ice and collecting over 80% of the outflow from the grounded ice, in its vast majority from the West Antarctic Ice Sheet (WAIS; Fowler and Ng, 2021).

Subshelf melting and calving are the main mechanisms responsible for ice-mass loss of the AIS today (Depoorter et al., 2013). Thus, the WAIS, which is broadly a marine-based ice sheet, experiences the highest melt rates through the critical influence of warmer water in contact with the ice, especially in the Amundsen and Bellinghausen Seas (Rignot et al., 2013).

Enhanced basal melting leads to ice-shelf thinning. Since ice shelves are floating, this does not lead to an increase in sea level per se. However, laterally-confined ice shelves buttress the upstream flow. Therefore their thinning can reduce their buttressing effect, accelerating inland ice flow and discharge, thereby contributing to sea-level increase (Fürst et al., 2016).

Several areas of the AIS have experienced significant ice mass loss in the past decades, contributing approximately 8 mm to the sea level rise since 1992 (Bell and Seroussi, 2020 and references therein). Notably, most ice mass loss has originated in the WAIS. The ice loss in this region has increased by 70% in the last decades (Paolo et al., 2015), ejecting approximately 3331 Gt of ice to the Amundsen Sea (Davison et al., 2023). This is explained by the reinforced output of some major nearby glaciers such as Pine Island, which has sped-up and reached its maximum velocity at $\sim 4000$ m/yr in 2009 (Joughin et al., 2021), and Thwaites Glacier, which is responsible for 15% of the WAIS ice loss (Schmidt et al., 2023).

An increase in the discharge of ice across the grounding line causes it to retreat. If the ice sheet is grounded on a retrograde slope, this retreat is thought to lead to an intrinsic instability, the marine ice-sheet instability (MISI; Schoof, 2007; Weertman, 1974), where a positive feedback is triggered. On a retrograde slope, the retreat of the grounding line implies an increase in the ice thickness in that area. This enhances the discharge of ice across the grounding line, which drives a further retreat (Pattyn and Morlighem, 2020). The MISI is an important source of uncertainty in future projections of the AIS (Robel et al., 2019). Several studies have suggested that the aforementioned sectors of the WAIS that are experiencing fast ice-mass loss could be already undergoing MISI (Joughin et al., 2014; Rignot et al., 2013). The current stability of the WAIS has been recently studied by Hill et al. (2023), concluding that present-day retreat of Antarctic grounding lines is not yet irreversible or self-sustained. Nevertheless, in a continuation of this work, Reese et al. (2022) showed that irreversibility could appear under present-day climate forcing if the grounding lines retreated further inland.

In the future, under global warming, there is the risk of additional mass loss from the AIS both under sustained warming of the Southern Ocean and the increased tendency to produce upwelling of Circumpolar Deep Waters (CDW) caused by stronger westerly winds (Holland et al., 2020, 2022). Complete disappearance of the WAIS could lead to an estimated sea-level rise of 1.91-5.08 m (Sun et al., 2020). Nonetheless, there is substantial uncertainty in the future contribution of the AIS to sea-level rise (Edwards et al., 2021; Seroussi et al., 2020), which ranges from only about $\sim 10$ cm (Ritz et al., 2015) to above 1 m (DeConto and Pollard, 2016) in this century. Actually, as stated in the latest IPCC report (Masson-Delmotte et al., 2021b), the AIS is the largest source of uncertainty in multi-centennial sea-level projections, with a potential contribution by 2300 ranging from -0.14 m to 0.78 m in low emissions scenarios and from -0.27 m to more than 3 m in high emissions scenarios (IPCC 2021, Masson-Delmotte et al., 2021a). The main reason for this is the large uncertainty in the projections of ice-shelf stability in warming scenarios (van de Wal et al., 2022), which depends critically both on the climate forcing and on ice-ocean interaction processes that are not well constrained. Observations of large-scale ice sheets and cavities under ice shelves, where basal melting takes place, are scarce and thus our understanding of the interactions in these places is limited (Jourdain et al., 2020; Schmidt et al., 2023). Accordingly, melt rates generated by the ocean, their calibration based on oceanic conditions taken outside of ice shelf cavities, and the ice-sheet dynamic response to these oceanic changes are deeply uncertain (Seroussi et al., 2020).

Quantifying this uncertainty is a difficult task. Ideally, the proper tool to model ice-ocean interactions are coupled ice-ocean models (Gladstone et al., 2020; Park et al., 2023). However these models are computationally expensive and therefore their use for continental-scale ice-sheet studies is rare. An alternative to study the future evolution of the AIS is to use an offline approach by forcing an ice-sheet model with prescribed ocean temperatures, using a basal melt parameterization (e.g., Jourdain et al., 2020). Following this approach, Favier et al. (2019) analyzed the performance of a variety of basal melting laws. Different models use different parameterizations and therefore can lead to different responses of the AIS under the same forcings. The simplest law considers a linear dependency on thermal forcing with constant heat exchange velocity controlling the rate of melting. A further step in complexity is using a quadratic dependency on the thermal forcing. This allows capturing the effect by which melting induces stronger circulation under the shelves with warmer ocean conditions, fostering more melting, resulting in a positive feedback. In the simplest cases, sub-shelf melting parameterizations are local in the sense that basal melting on each point of the ice shelf only depends on the physical properties at that point. Jourdain et al. (2020) developed an extension of the quadratic law that takes into account non-local processes. This quadratic non-local parameterization was the standard parameterization used in the 6th Phase of the Ice-Sheet Modelling Intercomparison Project for CMIP6 (ISMIP6, Jourdain et al., 2020, Nowicki et al., 2020, Seroussi et al., 2020). Using 13 ice-sheets models from different international groups, Seroussi et al. (2020) investigated the evolution of the AIS during the period 2015–2100, showing a wide spread in their results in terms of sea level, covering an interval between -7.8 and 30 cm under the RCP8.5 scenario, with the WAIS adding roughly 18 cm and the East Antarctic Ice Sheet (EAIS) acting with a counterbalance effect gaining ice mass. These results suggest that if the divergence between models and oceanic conditions is significant by 2100, it would be expected to be even greater on longer time-scale simulations.

In fact, as noted by Lowry et al. (2021), the effects of global warming over Antarctica will have consequences in the long term, beyond the 21st century. Hence, an extension of the range period covering the projections in the ISMIP6 project is needed. A few studies have performed such extensions, analyzing different sources of uncertainty to shed light on what could happen in the coming centuries. Precisely, using a statistical emulator and the NorESM1-M model as forcing, Lowry et al. (2021) sampled several parameters relevant for ice-sheet dynamics (the sliding law exponent, the minimum till friction angle and enhancement factors) to estimate the probability distribution of the AIS contribution to sea level by 2300 under the RCP2.6 and RCP8.5 scenarios. The width of the spread in sea-level projections was found to broaden as time surpasses 2100, with an amplitude of several meters in 2300. Bulthuis et al. (2019) showed that the basal sliding law parameters, calving and ocean melt factors are all major sources of uncertainty for projections by the year 3000, with MISI ruling in marine basins. Chambers et al. (2021) also extend their projections until 3000, considering a constant climate after 2100 using different GCMs and ocean conditions, and demonstrating also that uncertainty is inherent to projections. Lipscomb et al. (2021) and Berdahl et al. (2023) made extensions until 2500 (see details in Section 4), but considering a repeated forcing using the two-decadal range 2081-2100 in order to account for climate variability, and an averaged climatology of the period 2090-2100, respectively, with the original ISMIP6-2100 dataset. This leads to an underestimation of the thermal forcing for the period 2100-2500 (see Table 5). More recently, Seroussi et al. (2024, under review) considered extensions until 2300 with new GCMs forcings ranging until that same year.

Here we investigate the sensitivity of our ice-sheet model to the three main sources of uncertainty: future scenarios, climate-model forcings and ice-ocean interactions in stand-alone ice-sheet model simulations by extending the ISMIP6 projections beyond the 21st century, until 2500 with a similar procedure to Lipscomb et al. (2021) and Berdahl et al. (2023), but using for our purpose, a subset of the 14-member ensemble of experiments given by the protocol for ISMIP6-2300 (Jourdain et al., 2020, Nowicki et al., 2020, Seroussi et al., 2020, Seroussi et al., 2024, under review), taking GCMs from the CMIP5 and CMIP6 ensembles under different scenarios until 2300 extended to 2500 and applying the quadratic non-local parameterization for basal melting in order to study the sensitivity in ice-ocean interactions. Atmospheric and oceanic forcings are also considered independently in order to detach their effects over the AIS and potentially identify the main driver of the changes on the ice sheet. Furthermore, we also focus on the timing of the loss of ice in the WAIS in the different simulations. This work is structured as follows: in Section 2 the ice-sheet model used, the basal melting parameterization chosen and the setup configuration for the simulations are described, together with the details of the experiments carried out. In Section 3, the results of the experiments are described and particularly analyzed in terms of the sea-level contribution with respect to a control simulation. In Section 4 we discuss our results in the context of previous research studies and the outlook for future work, and finally in Section 5 the main conclusions are summarized.

## 2 Methodology

In this section the ice-sheet model and the basal melting law used in the study together with the experimental setup are described.

### 2.1 Description of the ice-sheet model

The ice-sheet model used in this study is Yelmo (Robinson et al., 2020). It is a 3D thermomechanical model containing four main components: topography, material properties, dynamics and thermodynamics. Each of these components correspond to the calculation of specific state variables. Nevertheless, due to the fact that the equations involving the dynamics in ice are nonlinear and coupled with other parts of the model, these components are in turn tightly coupled with each other internally. Yelmo is a higher-order model, in other words, it uses methods that take into account longitudinal, lateral and vertical shear stresses. Recently, it has incorporated one of these methods, known as the depth-integrated effective viscosity approximation (DIVA), which permits solving the horizontal momentum balance in 2D while maintaining high fidelity to the full Stokes solution and running fast at higher resolutions (Goldberg, 2011; Robinson et al., 2022). Yelmo has been tested in benchmark experiments such as EISMINT1, EISMINT2 and MISMIP (Robinson et al., 2020). These idealized tests give confidence that the model performs well for known conditions. It has also recently been used to simulate the AIS (Blasco et al., 2021) and the Laurentide Ice Sheet at the Last Glacial Maximum (Moreno-Parada et al., 2023).

Regarding resolution, a regular grid composed of 381 x 381 cells for a horizontal resolution of 16 km and 10 vertical layers has been used for this study. Nonetheless, additional tests with 32-km cells have been made for comparison (see Section 4).

In this study, basal stress $\tau_b$ was implemented as a regularized Coulomb power law (Joughin et al., 2019; Schoof, 2005), given in terms of the basal velocity $\mathbf{u}_b = (u_b, v_b)$ as:

$$\tau_b = -c_f N \left( \frac{|\mathbf{u}_b|}{|\mathbf{u}_b| + u_0} \right)^q \frac{\mathbf{u}_b}{|\mathbf{u}_b|} \tag{1}$$

where $N$ is the effective pressure produced by the ice at the base, $q = 0.2$ an exponent and $u_0 = 100$ m/yr an empirical threshold speed (Zoet and Iverson, 2020). $c_f$ is a tunable dimensionless parameter that represents the frictional properties of the bedrock (see Section 2.3). For the effective pressure $N$, the parameterization by Leguy et al. (2014) is selected

$$N(p) = \rho_i g H \left( 1 - \frac{H_f}{H} \right)^p \tag{2}$$

with $g$ being the acceleration due to gravity and $H_f$ the flotation thickness (dependent on $\rho_{sw}$ and bed elevation). The value of the parameter on the exponent $p$ represents the hydrological connectivity of the subglacial drainage system to the ocean and is set to 0.5.

Concerning calving, a von Mises calving law (Lipscomb et al., 2019) is used with the scaling calving parameter $k_t = 0.0025$ m yr$^{-1}$Pa$^{-1}$ and $w_2 = 25$ for the calving eigenvalue weighting coefficient, as in the approach by (Lipscomb et al., 2019). For the geothermal heat flow, the reconstruction of Shapiro and Ritzwoller (2004) is used. Furthermore, ELRA is used as GIA model, but in a five-century period the bedrock is not expected to change considerably. Nonetheless, Seroussi et al. (2024, under review) have demonstrated (pers. comm.) that an ELRA model correction in the bedrock elevation could result in differences of around +11-17% per $\sim$1.3 m SLE calculated with the volume above flotation method, which should be kept in mind, as we have used this conversion in our study. As Goelzer et al. (2020) pointed out, on short timescales, this conversion can assume a constant area for the ocean when calculating the sea-level contribution.

## 2.2 Basal melting parameterization

Several studies have suggested that a quadratic melt parameterization is best able to mimic the ice–ocean thermal forcing coupling as compared with other basal melting laws (Seroussi et al., 2020; Burgard et al., 2022). For this reason, the generalized non-local basal melt parameterization described in Jourdain et al. (2020) is chosen in this study. This parameterization takes into account not only the local forcing, but also the average forcing over the cavity beneath a given ice shelf. It translates through $\gamma_0$, the parameter describing the rate at which heat is transferred between ice and ocean, the thermal forcing into basal melting (of floating ice), $B_f(x, y)$ by:

$$B_f(x,y) = \gamma_0 \times \left( \frac{\rho_{sw} c_{pw}}{\rho_i L_f} \right)^2 \times (\mathrm{T_F}(x, y, z_{\text{draft}}) + \delta T_{\text{sector}}) \times |\langle \mathrm{T_F} \rangle_{\text{draft} \in \text{sector}} + \delta T_{\text{sector}}| \tag{3}$$

where it has been denoted by $\mathrm{T_F}(x, y, z_{\text{draft}})$ the thermal forcing at the ice-ocean interface with $z_{\text{draft}}$ the thickness of the ice shelf, $\rho_{sw} = 1028$ kg m$^{-3}$ is the density of ocean water, $\rho_i = 918$ kg m$^{-3}$ the density of ice, $L_f = 3.34 \times 10^5$ J kg$^{-1}$ the latent heat of fusion of ice, $c_{pw} = 3974$ J kg$^{-1}$K$^{-1}$ the specific heat of the ocean water and $\delta T_{\text{sector}}$ is a temperature correction (thermal forcing correction). The term $\langle \mathrm{T_F} \rangle_{\text{draft} \in \text{sector}}$ represents the thermal forcing averaged over all ice shelves contained in

each of the 18 drainage basins in which the Antarctic domain is divided (Jourdain et al., 2020). Yelmo includes a marine-shelf interface that allows to interpolate the thermal forcing at the required depths. First, it computes the depth of the ice shelf base by distinguishing between floating and grounded ice. Second, the weights for the different vertical layers are calculated by taking into account the two nearest layers of the depth of the shelf previously calculated in the first step. Finally, the field is computed in terms of these weights as a weighted sum.

Finally, concerning sub-shelf melting in the vicinity of the grounding line, there is a debate as to how strong the latter should be. In the literature two common options are the no melt parameterization (NMP) and the partial melt parameterization (PMP) (Leguy et al., 2021; Seroussi and Morlighem, 2018). Here, the PMP parameterization is used. In Yelmo, each individual cell of the domain has assigned values between 0 and 1 for the fraction of the area of the cell that is floating $\phi_f$ and grounded $\phi_g$ in such a way that $\phi_g = 1 - \phi_f$. Given a value $m$ for basal melting in a partially-grounded cell, the PMP parameterization uses $\phi_f$ as a weighting factor for $m$.

## 2.3 Experimental setup

A subset of five future-projection experiments defined in the protocol for ISMIP6-2300 (Jourdain et al., 2020, Nowicki et al., 2020, Seroussi et al., 2020, Seroussi et al., 2024, under review) is performed here (Table 1), namely the extended scenarios to 2300 without repeated forcing or ice-shelf collapse. These experiments use different AOGCM models from CMIP5 (CCSM4 and HadGEM2) and CMIP6 (CESM2 and UKESM1), Table 1. Regarding the scenario, higher emissions are used in four of the experiments leaving only one experiment with a low emission scenario. In this way, we focus on the structural uncertainties produced by higher emissions. Although in the protocol the ending year is established in 2300, in this study we extend the simulations towards 2500, by forcing from 2300 to 2500 with the average of the last 10-year period (2291-2300) of the experiments in Table 1.

In ISMIP6-2300, there are four experiments considering global warming conditions (from expAE02 to expAE05 in Table 1). Although their atmospheric forcing anomaly shows a similar evolution in time, ranging approximately between 10 and 15 K (Fig. 1a), there is a mismatch in the oceanic forcing observed in CCSM4, which is between 1 and 2 K below the other higher-emission curves. Therefore, the oceanic thermal forcing has an independent reaction compared to the atmospheric forcing. For instance, while UKESM1 and CCSM4 have very similar atmospheric forcings, reaching the end of the period with nearly the same surface temperature anomalies, their oceanic thermal forcings are very different, diverging at the end of the period by nearly 2 K (Fig. 1b).

The contrast between the forcings can also be seen in Fig. 1, for the two main regions of the AIS. For surface temperature anomaly (Fig. 1d), there is no difference in the evolution of the higher-emission scenarios. Meanwhile, for the oceanic forcing some differences appear, both for fixed (present-day) and evolving grounding lines (Figs. 1e and 1f). With the evolving grounding lines, the WAIS is between 0.25 and 0.75 K warmer than the EAIS on average (Fig. 1f).

High-emission scenarios lead to higher surface temperatures all over Antarctica (Fig. 2a). Regions particularly affected are the Ross ice shelf and the Amundsen Sea, with temperatures rising more than 15 K in 2300 with respect to the present day. Also, surface temperature anomalies in the interior of the AIS in CESM2-WACCM surpass 25 K. The low emission case,

however, experiences a cooling in some regions, especially in the waters off Victoria Land, the Amundsen and the Weddell seas. Similar differences in thermal forcing are observed in ice-shelf cavities (Fig. 2b), with increases of nearly 8 K for CESM2 and UKESM1-0-LL in the WAIS for Ronne-Filchner and Ross, but also underneath the EAIS ice shelves (Amery, Shackleton and the Wilkes Land). These changes can also be observed in terms of SMB anomaly where in the WAIS, especially the Antarctic Peninsula, and the EAIS predominate the accumulation of ice in the cases of CESM2-WACCM and UKESM1-0-LL with high emissions (Fig. A2).

For each forcing scenario, a range of values for the heat exchange velocity, $\gamma_0$, is considered in the quadratic non-local basal melting parameterization (Eq. 3) by taking into account the distribution based on circum-Antarctic observations associated with the MeanAnt calibration, proposed by Jourdain et al. (2020) and used in the context of ISMIP6 by Seroussi et al. (2020). The 5th, 50th and 95th percentiles are used yielding an interval ranging from a low to a high value for $\gamma_0$ (Table 2). In this way, a 3-member ensemble is obtained for each forcing scenario.

For the initialization of the projections, spin-up simulations running for 20 kyr with a constant reference forcing were carried out. For the atmospheric fields, Surface Mass Balance (SMB) and 2m-air temperature, we have used monthly values from the Regional Atmospheric Climate Model (RACMO v2.3, Van Wessem et al., 2014) forced by ERA-Interim reanalysis (Dee et al., 2011) averaged over the time period 1981-2010. Regarding the ocean, the fields produced by Jourdain et al. (2020) for ISMIP6 are used. To evaluate the sensitivity with respect to the heat exchange parameter $\gamma_0$, a different spin-up is required for each experiment. During the first 15 kyr of the spin-up, the spatially explicit basal friction coefficient $c_f$ (Eq. 1) is obtained through an optimization process so that the modeled ice thickness $H$ is as close as possible to the observed present-day state $H_{obs}$. The observed variables are taken from the BedMachine Antarctica V2 dataset (Morlighem et al., 2020) and they also represent the initial configuration of the ice sheet before the spin-up is performed. To optimize $c_f$, the time-evolution differential equation from Lipscomb et al. (2021) is considered:

$$\frac{dc_f}{dt} = -\frac{c_f}{H_0}\left[\frac{H - H_{obs}}{\tau_c} + 2\frac{dH}{dt} + \frac{H_0}{20}\frac{\log(c_f/c_{f,\text{target}})}{\tau_c}\right] \tag{4}$$

where $H_0 = 100$ m and $\tau_c = 500$ years are constants for scaling in ice thickness and relaxation time respectively, and $c_{f,\text{target}}$ is a target for $c_f$. The last term ensures that in places where the optimization is less successful, the value of $c_f$ does not saturate to an extreme value (William H. Lipscomb, pers. comm.). $c_{f,\text{target}}$ is defined as a function of elevation following Winkelmann et al. (2011). In order to compute the thermal forcing correction, $\delta T_{\text{sector}}$ in Eq. 3, an equivalent optimization process to the one described for the basal stress constant $c_f$ is carried out. A minimum limit was set at -3 K and a maximum at 0.5 K, so that $\delta T_{\text{sector}}$ values on each sector are consistent with present-day conditions. As pointed out by Lipscomb et al. (2021), this is a different perspective to that developed in Jourdain et al. (2020) who consider Antarctica's basins with fixed thermal forcing values instead. Additional analysis as to how these two optimised parameters affect the initialization of the experiments can be found in Section 4. For the last 5 kyr of the spin-up, the optimised fields are held constant and the simulation is run with the reference climatology towards steady state.

A control simulation using the same climate forcing as its spin-up simulation is carried out starting in 2015 and running until 2500. The only difference with respect to the spin-up run is the fact that optimization is not used for neither of the two

**Table 1.** Description of the five experiments carried out based on the protocol for ISMIP6-2300. The ending year of the forcing in these experiments is 2300 and ice-shelf collapse is not imposed. Note that in this study we extend the simulations towards 2500 by using the average of the last 10-year period (2291-2300) of the experiments.

| Experiment | Model | Scenario |
|---|---|---|
| ctrlAE | — | — |
| expAE02 | CCSM4 | RCP8.5 |
| expAE03 | HadGEM2(-ES) | RCP8.5 |
| expAE04 | CESM2(-WACCM) | SSP5-8.5 |
| expAE05 | UKESM1(-0-LL) | SSP5-8.5 |
| expAE10 | UKESM1(-0-LL) | SSP1-2.6 |

**Table 2.** Reference names and values of $\gamma_0$ used in the experiments. The values are chosen considering the MeanAnt calibration methodology from Jourdain et al. (2020).

| Reference name | Percentile | $\gamma_0$ (m/yr) |
|---|---|---|
| Low | 5th | 9620 |
| Medium | 50th | 14500 |
| High | 95th | 21000 |

variables mentioned before. For each value of $\gamma_0$, a separate spin-up and control simulation is performed. The control runs with the different values of $\gamma_0$ are all stable during the simulation, with variations of about 5 mm of SLE between 2015 and 2500 in terms of volume above flotation in the three cases. Nevertheless, with increasing values of $\gamma_0$, the ice sheet volume decreases
at its initial state (Fig. A3).

### 2.4 Individual contribution of the atmosphere and the ocean

In order to study the separate effects of the atmospheric and oceanic forcings on the AIS, additional experiments were conducted: atmosphere-only runs in which the constant oceanic field at 2015 for thermal forcing is kept constant during the whole period of simulation and ocean-only simulations with the SMB and surface temperature fields at 2015 held until 2500.

### 2.5 Error analysis in the initialization procedure

In Figure 3, the root mean square error (RMSE) has been computed for ice thickness and ice-surface velocity at the beginning of the control run with respect to the observations given by the BedMachine Antarctica V2 dataset (Morlighem et al., 2020) and the Rignot et al. (2011) dataset, respectively. This RMSE has been computed using two masks, one with the initial configuration

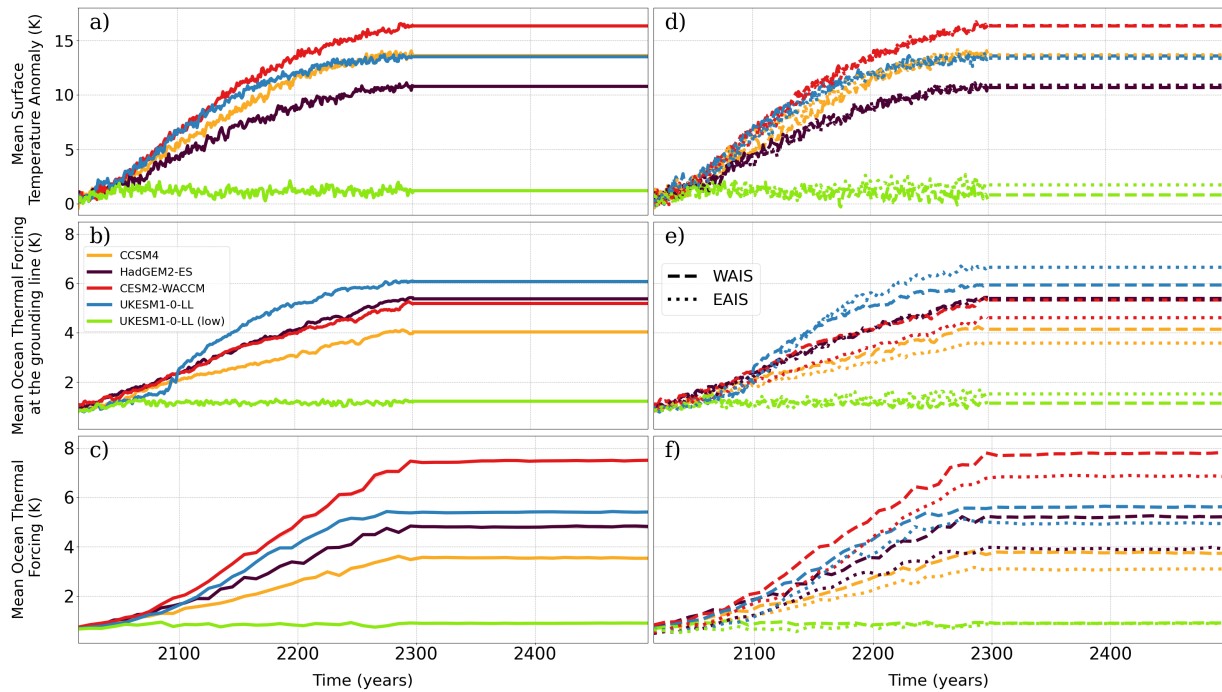

**Figure 1.** Mean evolution of the atmospheric (a), oceanic thermal forcing at the grounding line depth fixed in 2015 (b) and thermal forcing at the evolving grounding lines in the AIS (c) for the five experiments carried out: CCSM4 (RCP8.5), HadGEM2-ES (RCP8.5), CESM2-WACCM (SSP5-8.5), UKESM1-0-LL (SSP5-8.5) and UKESM1-0-LL (SSP1-2.6). For (d), (e) and (f), same as in (a), (b) and (c) respectively but for the WAIS and EAIS. From 2300 to 2500 a constant climate is imposed through the average climatic conditions in the period 2291-2300.

of the AIS given by our initialization where ice shelves are allowed to expand further than observations, and the other with
the observations. In the first case, for ice thickness, the deviation from observations is in the range 201 to 216 m, decreasing for increasing $\gamma_0$. For ice surface velocity, the corresponding range is between 201 and 275 m/yr. Overall, Yelmo's root-mean square errors for ice thickness and ice surface velocity are in the medium and upper range, respectively, of the values obtained in ISMIP6 (see Figure 3 in Seroussi et al., 2020). For the medium value of $\gamma_0$ the model overestimates ice thickness, most notably in the ice-sheet margins (Fig. 4a). Only a few regions in the western (Fig. 4c) and eastern sectors show thinner ice
than observations. There is a similar pattern in surface velocity anomalies (Fig. 4b), with velocities on the ice shelves generally overestimated. Yet, in the Ronne-Filchner, Ross and Pine Island ice shelves regions, there are a few ice streams showing lower ice surface velocities than observed (Fig. 4d).

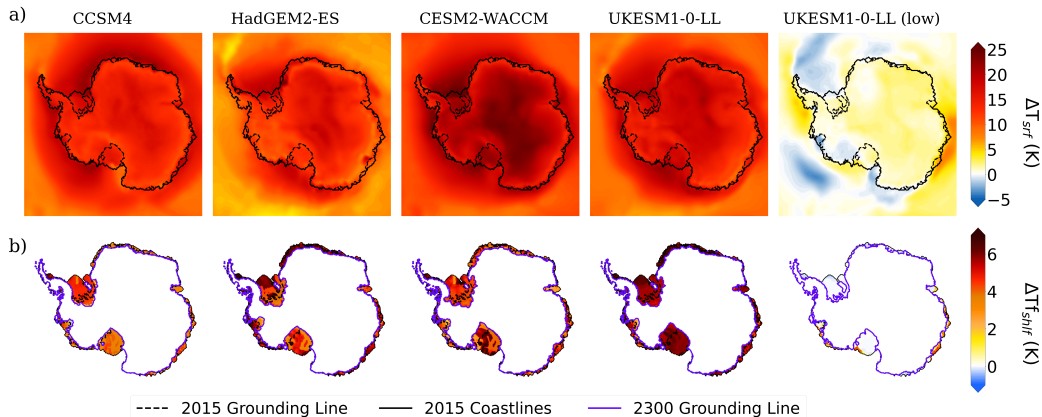

**Figure 2.** Surface temperature anomaly (a, in K) between 2300 and the present day as simulated by each of the GCMs selected from ISMIP6-2300 in Table 1, using four high-emission scenarios and a low emission one. Thermal forcing differences over the ice shelves (b, in K) between 2300 and 2015, together with the grounding lines and coastlines at the initial time and the 2300 grounding line.

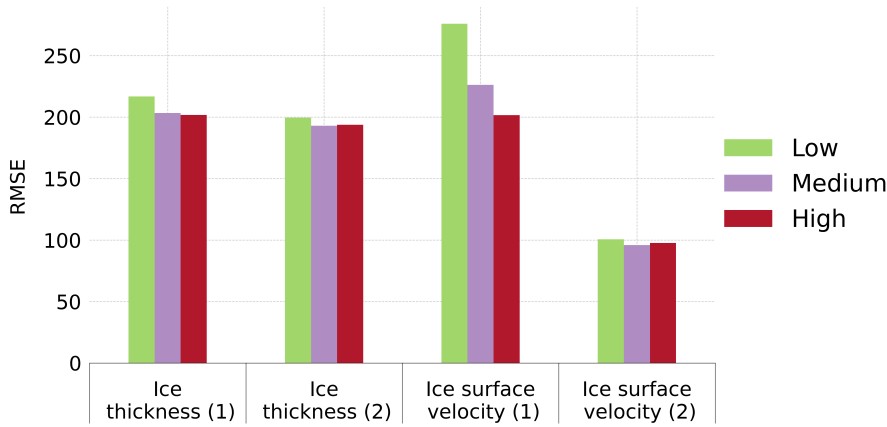

**Figure 3.** Root mean square error (RMSE) for ice thickness (in m) and ice surface velocity (in m/yr) between the start of the simulations in 2015 and observations. The different colors represent the values of $\gamma_0$. To mask over the cells containing ice, the ice-sheet margins at the start of the simulation (extended ice shelves) have been used for the columns labelled with (1) and the observations for the ones labelled with (2).

## 3 Results

### 3.1 Sensitivity to climate forcing and $\gamma_0$ value

The evolution of the contribution of the AIS to global sea level following the five projections derived from the experiments in Table 1 is shown in Figure 5, together with the spread produced by varying $\gamma_0$. CESM2-WACCM shows the highest long-term contribution to sea level, which is a consequence of this model leading to the maximum forcing both by the atmosphere and the ocean (Fig. 1). For this particular experiment, the medium value of $\gamma_0$ yields ca. 3.5 m SLE at 2500, and the difference between

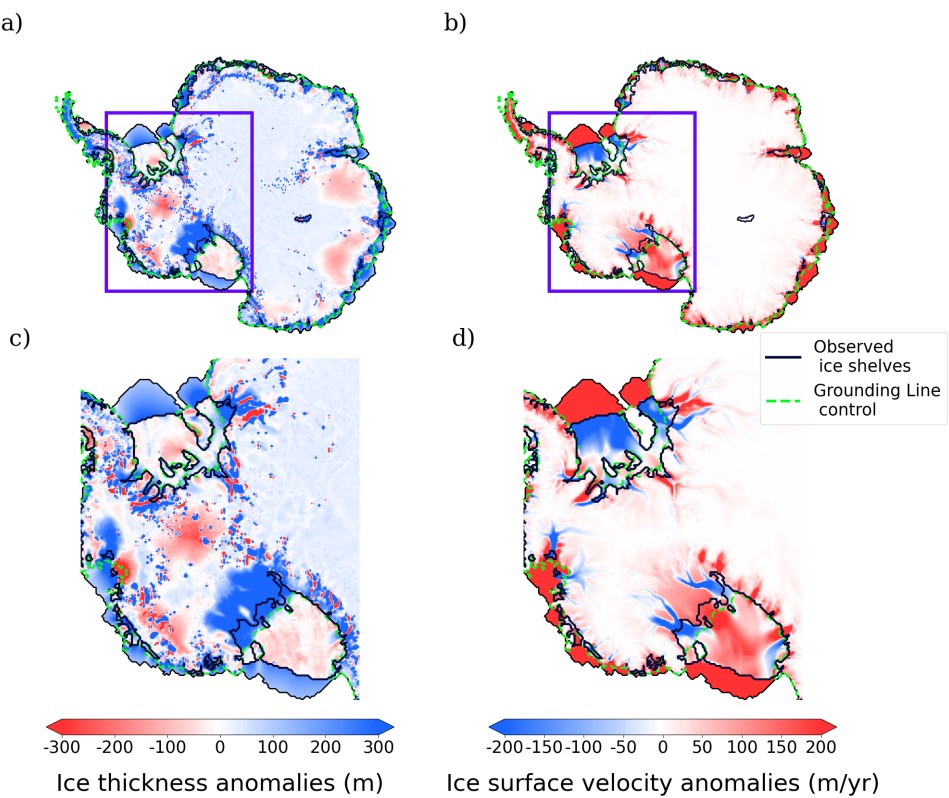

**Figure 4.** Ice thickness (a, in m) and surface velocity anomalies (b, in m/yr) at 2015 for the medium value of $\gamma_0$, with respect to observations from the BedMachine Antarctica V2 dataset and Rignot et al. (2011) dataset, respectively. Zoom over the WAIS region (c and d) to better appreciate the details at the grounding line.

using a low or high $\gamma_0$ value is close to 1.3 m SLE. Although CCSM4 and the high-emission case for UKESM1-0-LL have

similar atmospheric forcings, their contributions by 2500 differ by more than 2 m SLE. In fact, the bounds of the projections given by the different values of $\gamma_0$ do not overlap between these two experiments and furthermore the low-emission case is comparable with the CCSM4-driven experiments in the high-emission case. This is explained by the very different oceanic forcing under the AIS ice shelves, as demonstrated by the diverging curves of CCSM4 and the high scenario for UKESM1-0-LL in Figs. 1b and 1c, despite having very similar atmospheric forcing time series. In fact, in the period 2015-2100 the mean

thermal forcing at the grounding line increments $\sim 1$ K for all high-emission scenarios, with CCSM4 and UKESM1-0-LL showing roughly 2 K at 2100. However, between 2100 and 2300, the thermal forcing rate with CCSM4 is approximately 1 K per century, while UKESM1-0-LL doubles that rate. This points to the critical role of the ocean forcing.

Considering the extension from 2300 to 2500 under constant climate forcing, we can see the sea-level contributions slow down in the last two centuries (Fig. 5b). For the medium value of $\gamma_0$, the rate of sea-level contribution for CESM2-WACCM,

HadGEM2-ES and UKESM1-0-LL (SSP5-8.5) reaches a peak near the year 2300 with values between 10 to 15 mm SLE/yr.

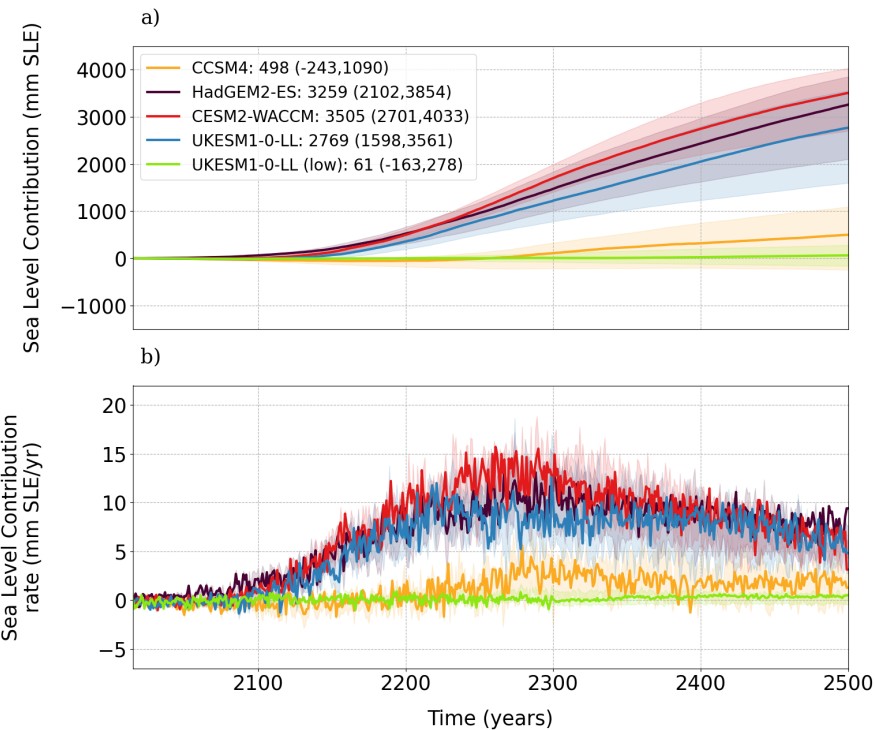

**Figure 5.** a) Projections of the AIS sea-level contribution (mm SLE) for the five scenarios chosen (Table 1), relative to the control run. In shaded colours the spread resulting from the range of $\gamma_0$ values is represented; the solid line corresponds to the medium value. The values in the legend are indicative of the sea-level contribution given by the medium value followed inside the parentheses by the corresponding low and high value. b) Same as a) but for the rate of sea-level contribution (mm SLE/yr).

After 2300, these rates go below 10 mm SLE/yr ending up with nearly 5 mm SLE/yr in 2500 with the exception of HadGEM2-ES, close to 10 mm SLE/yr, but no simulation reaches a new equilibrium before the year 2500 even though the forcings are kept constant after the year 2300.

A summary of the results at 2500 in terms of sea level can be found in Figure 6 for each value of the parameter $\gamma_0$, where the effect of increasing basal melting is reflected.

### 3.1.1   Sensitivity analysis on the WAIS and EAIS

On a regional basis, the WAIS (including the ice sheet over the Antarctic Peninsula, APIS) is the main contributor to the global sea-level rise (Fig. 7), with all simulations showing positive contributions for the medium value of $\gamma_0$ and reaching up to 2 m in the cases of HadGEM2-ES, CESM2-WACCM and UKESM1-0-LL. For these three forcings, the EAIS also shows

a positive contribution to sea level but never exceeding $\sim 1$ m SLE. The relative contributions of the WAIS (74-80%) and the EAIS (20-26%) are similar across these three models (Table 3), with the WAIS being the major contributor until the year 2500.

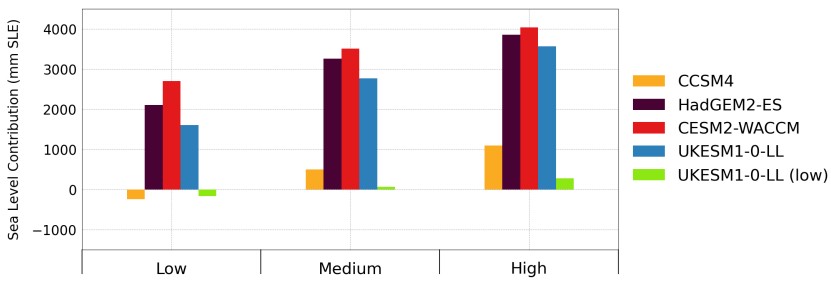

**Figure 6.** Bar plots for the sea-level contribution (mm SLE) at 2500 for the different GCM forcings from ISMIP6-2300.

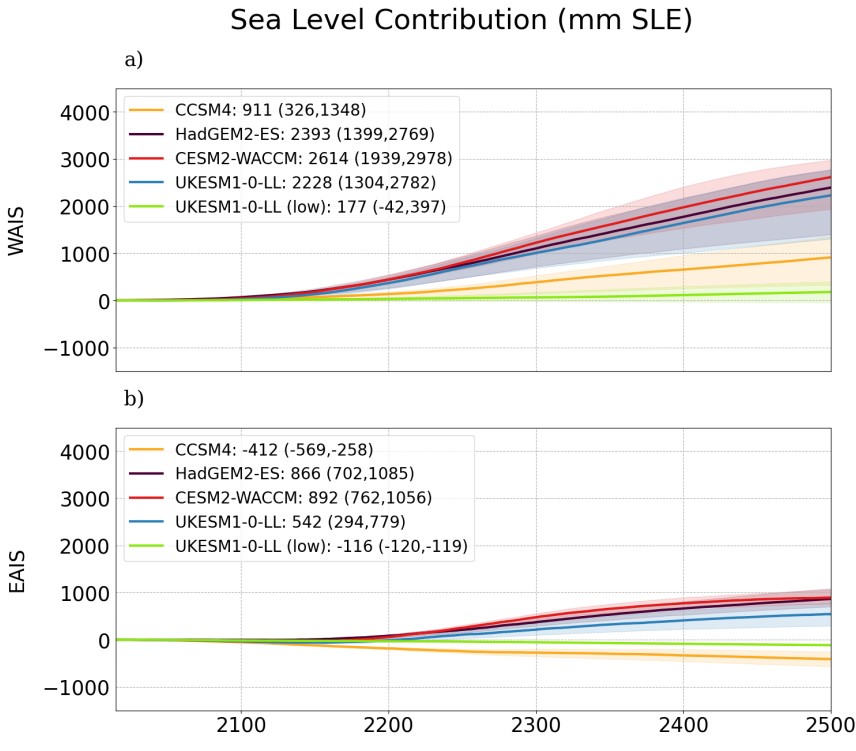

**Figure 7.** Evolution of the sea-level contribution (mm SLE) from the two main regions of Antarctica, the WAIS (including the Antarctic Peninsula, a) and the EAIS (b), relative to the control run, since the start of the simulations in 2015 to year 2500.

Meanwhile, for CCSM4 and UKESM1-0-LL (low), the EAIS counterbalances the contribution of the WAIS by lowering sea level through a gain in accumulation.

**Table 3.** Total contribution (in mm SLE) and percentage of contribution of the WAIS and the EAIS to the total amount of the AIS volume variations at year 2500 with the five different climate forcings using the medium value of $\gamma_0$ in Table 2.

|  | CCSM4 | HadGEM2-ES | CESM2-WACCM | UKESM1-0-LL | UKESM1-0-LL (low) |
|---|---|---|---|---|---|
| Total Contribution (mm SLE) | 498 | 3259 | 3505 | 2769 | 61 |
| WAIS (%) | 182 | 74 | 74 | 80 | 290 |
| EAIS (%) | -82 | 26 | 26 | 20 | -190 |

## 3.2 Spatial patterns in the loss of ice

To gain insight into the reasons behind the behaviour of the time series shown, it is useful to compare the ice thickness and ice surface velocity anomalies at the beginning of each century until 2500 with respect to the start of the projections in 2015 for the medium value of $\gamma_0$ (Figs. 8 and A4, respectively). The WAIS loses mass in all experiments under high-emission scenarios (Figs. 8 and A5). This can be explained by the warmer waters in the Amundsen and Bellingshausen seas sectors (Fig. 2). In addition, as shown in Figure 8, the EAIS also loses ice mass, and the Amery ice shelf serves as an example. The loss of ice

takes place particularly in the margins of the ice sheet, while there is gain of ice in the interior, especially in the eastern sector. At 2100, the experiments with CESM2-WACCM and CCSM4 show a clear loss of ice in the Ronne-Filchner and Amery ice shelves. However, for CCSM4 there is a delay in the loss of ice compared with the other higher emission experiments in the subsequent centuries. Changes in the position of the grounding line of the Ross ice shelf are not evident until 2400 for CCSM4, while for CESM2 the grounding line begins to retreat between 2200 and 2300. Despite the constant climate forcing over the

last two centuries, the grounding line keeps retreating in the WAIS from 2300 to 2500. The collapse of the ice shelves fringing the WAIS leads to debuttressing of the interior ice flow (Figs. A4 and A6). Hence nearly all of the grounded ice of the WAIS ends up disappearing in all the experiments with higher emissions with the exception of CCSM4.

The loss of ice in the WAIS, reaching anomalies in ice thickness well above 300 m, activates a retreat in the grounding lines, mainly in the Ronne-Filchner, Ross, Pine Island Glacier and Thwaites Glacier ice shelves, leading to a debuttressing effect that

amplifies ice flow and permits the ocean to advance towards the interior, until finally the grounding line migrates hundreds of kilometers, destabilizing the ice shelves and in turn producing even more loss of ice in those regions. This is in contrast with the low-emission scenario with UKESM1-0-LL, where the grounding line position remains largely unchanged and even the Ronne-Filchner ice shelf is larger, in the sense that is gaining ice with respect to the start of the simulations. However, Pine Island and the Ross ice shelf are losing mass even under a low-emission forcing. In connection with the loss of ice, there is

an acceleration of the ice-stream flow, with the exception of the Ross ice shelf under low emissions. Figure A4 shows how the ice surface velocity is enhanced with respect to 2015. Only some areas of the APIS for HadGEM2-ES, CESM2-WACCM and UKESM1-0-LL with high emissions present a clear slowdown, but for the rest of the AIS, the acceleration is pronounced, reaching anomalies of more than 100 m/yr compared with 2015 by 2200.

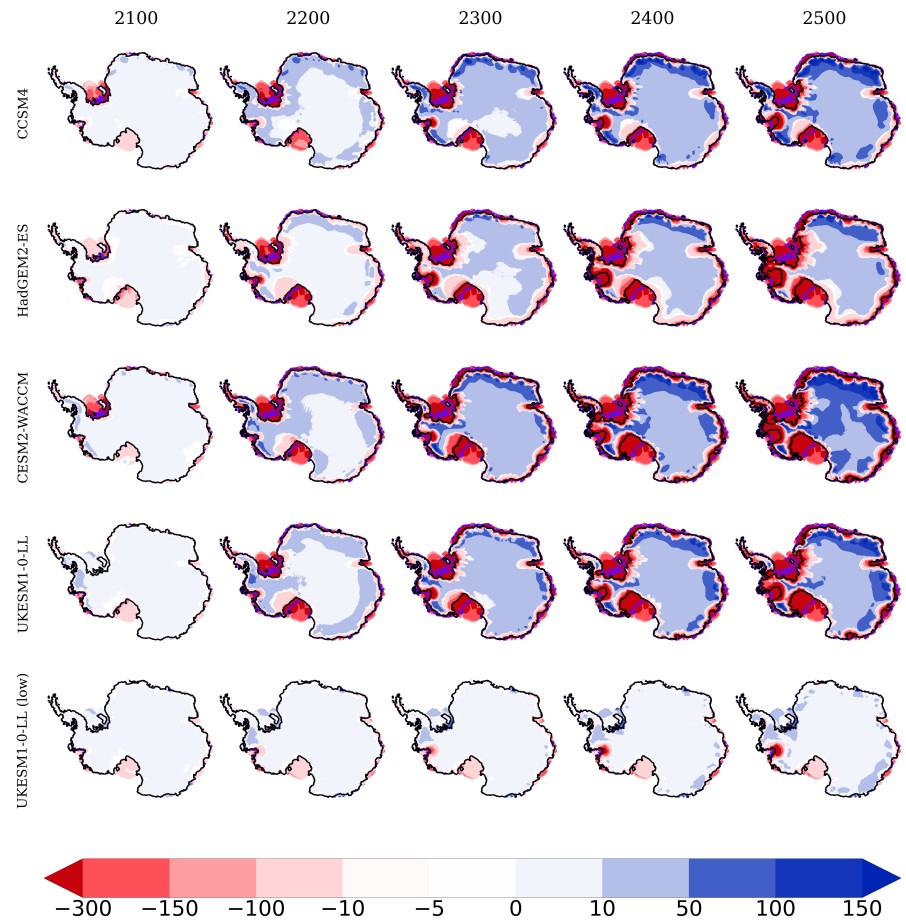

**Figure 8.** Ice thickness anomalies (in m) for the years 2100, 2200, 2300, 2400 and 2500 with respect to the start of the simulations for the set of five experiments carried out with the medium value of $\gamma_0$. The grounding line is represented by black and violet colours for 2500 and 2015 respectively. To show negative anomalies in the WAIS, no masking has been implemented as time evolves, keeping the original coastlines.

### 3.2.1 The timing of dynamic loss of ice in the WAIS

In order to investigate the sequence of events leading to the general loss of ice, in particular in the WAIS, we assess the evolution of the ice-area extension over this region (Fig. 9a). The total ice area shows a negative trend for all high-emission scenarios, with an abrupt change between 2100 and 2300 that leads to the loss of about 1 million $km^2$ in less than 100 years in the cases of HadGEM2-ES, CESM2-WACCM and UKESM1-0-LL. After the sharp ice-area reduction, the ice area continues to decrease at a lower rate. Nevertheless, Figure 9a also shows that this change is directly caused by the change in floating rather

than grounded ice, which evolves rather linearly. Yet, from 2300 onward, the floating ice area has almost disappeared,while the grounded area continues losing ice in all high scenarios.

The rate of change of ice volume above flotation in the WAIS is always negative (Fig. 9b). A minimum is reached near the end of the 23rd century. Hence, the rate at which the WAIS loses ice mass (and as a consequence its contribution to sea level) gradually increases until the year 2300, subsequently decreasing. This change in the tendency can be explained by the effect of the constant climatology imposed from 2300 to 2500 and also by the gain in accumulation produced by an increase in the precipitation in a warming climate (Masson-Delmotte et al., 2021b). For the EAIS (Fig. 9c), there is also a minimum near 2300 but the rate is much lower in absolute value.

As has been mentioned in Section 3.1, the experiments with CCSM4 show important differences with respect to the other high-emission simulations, in particular CESM2-WACCM, which is the more extreme example. Figure A7 shows the evolution of ice surface velocity for these two cases since 2100, where the WAIS begins to lose mass until 2300 where the contribution to sea level produced by this region changes its tendency (Fig. 9b). Both experiments differ drastically during the course of the 22nd century. For CESM2-WACCM, both the Ross and Ronne-Filchner ice shelves disappear while for CCSM4 only a retreat of these ice shelves towards the interior is observed. At 2300, once the forcing is set as constant for the remaining 200 years of study, the main shelves finally collapse for CCSM4, while for CESM2-WACCM, the grounding line continues migrating inwards reducing the area in the west (Fig. A7).

## 3.3  Individual effect of the atmosphere and the ocean

To assess the individual contributions of the atmosphere and ocean, the atmosphere-only and ocean-only runs are assessed. In the first case, the total contribution is less than 1.5 m SLE at the end of 2500 for all models (Fig. 10a). The simulations using UKESM1-0-LL, both with high and low emissions, and CCSM4 actually gain ice mass as a consequence of the enhanced accumulation rates (Fig. 11a) and the absence of ocean melting.

In contrast, for ocean-only runs, all experiments lead to a positive contribution to sea level (Fig. 10b). Uncertainties are roughly 2 m SLE between the low and high values of $\gamma_0$ for HadGEM2-ES, CESM2-WACCM and UKESM1-0-LL, with the medium values surpassing 3 m SLE by the end of 2500. For all simulations, the contribution to sea level from the ocean-only runs is above the corresponding run for the atmosphere-only case. Therefore, the ocean is clearly the main driver of the loss of ice and sea-level rise in Antarctica in all our projections. Figure 11a shows that for the atmosphere-only case, the WAIS loses ice as in the general case considering both forcings, but not enough to produce the collapse of the main ice-shelves. The EAIS ice shelves also have their ice mass reduced. In addition, in the interior and eastern areas, there is a gain of ice (Fig. 11a. Therefore, not considering the effects of the ocean allows the east to gain ice. Regarding the ocean-only case (Fig. 11b), the difference with the start of the simulations is found on the collapse of the main shelves in the WAIS and also in the EAIS. More interestingly, in the interior it is observed that changes are negligible, which can be explained by the fact that maintaining the present-day atmosphere does not lead to increased precipitation in the interior and eastern sectors, so no difference between 2015 and 2500 is found in terms of gain or loss of ice.

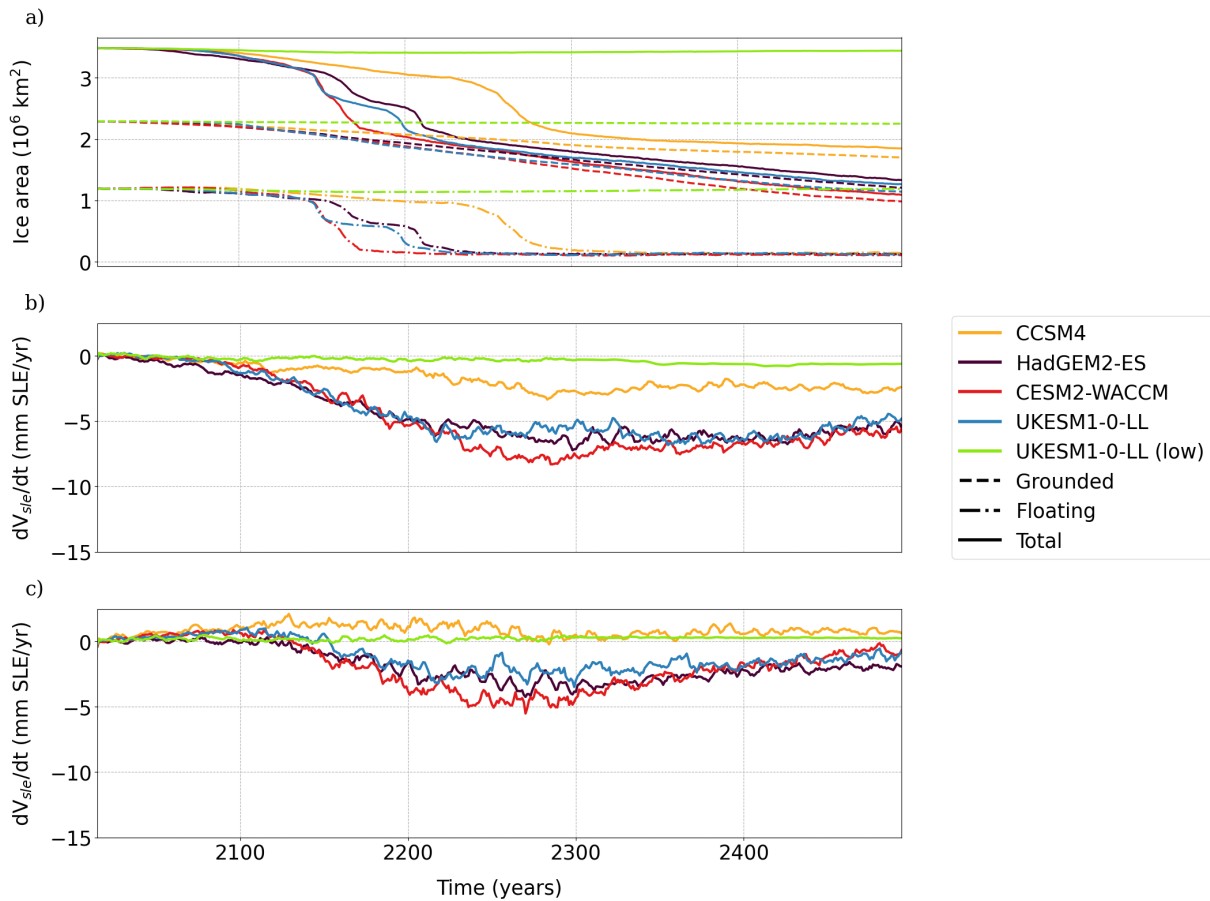

**Figure 9.** Ice area extension (a, in $10^6$ km$^2$) in the WAIS and rate of change in ice volume above flotation (in mm SLE/yr) in the WAIS (b) and EAIS (c) with time for the different GCM models considering the medium value of $\gamma_0$. In a), the grounded area is represented with dashed lines, the floating area with dash-dotted lines and the total area with solid lines.

## 4 Discussion

We have studied the sensitivity of the AIS to climatic forcing and scenarios on one hand and to the main basal melting parameter
on the other with the ice-sheet-shelf model Yelmo. To this end we have chosen an offline approach incorporating the ocean forcing through the basal melting parameterization of Jourdain et al. (2020), applying the PMP parameterization for melting at the grounding line. Based on our analysis of the RMSE for ice thickness and ice surface velocity, we can say that Yelmo is in the same range as other ISMIP6 models.

We have shown that the higher the value of the heat exchange parameter $\gamma_0$, the larger the future sea-level contribution.
For the medium value, the sea-level contribution varies from ~0.5 to 3.5 m SLE by 2500 under the effect of high-emission scenarios. The upper-range values correspond to the forcing given by CESM2-WACCM within the SSP5-85 scenario. More-

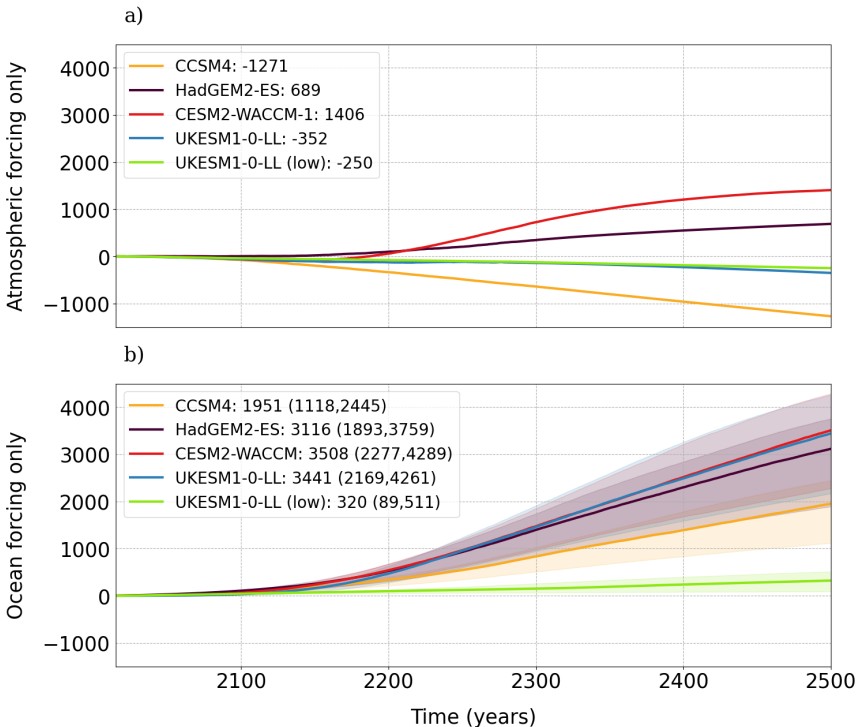

**Figure 10.** Sea-level contribution of the AIS (mm SLE) in the cases where only the forcing of the atmosphere (a) or the ocean (b) are considered. The shading in b) indicates the spread produced by the effect of the values of $\gamma_0$, while in a), as the optimization process leads to similar initial ice sheet states regardless of $\gamma_0$, the ocean fields maintain the present-day state.

over, we have shown that higher emission scenarios per se do not automatically lead to a higher sea-level contribution unless the ocean undergoes a strong warming, as illustrated by the simulations using the CCSM4 forcing with a RCP8.5 scenario. Furthermore, the rate of change of this contribution for higher emission scenarios increases until the end of the 23rd century, when the forcing is kept constant, but sea level continues to rise in the last two centuries at a lower speed.

The oceanic forcing is critical, and therefore it is important to represent it correctly. CMIP5 and CMIP6 future projections indicate that Antarctica's surrounding ocean will warm because of wind-driven circulation changes (they project a poleward wind shift of ∼0.8° and a strengthening of ∼10% under SSP585 by the end of this century), ocean warming far to the north of the coast, and enhanced intrusion of CDW into the cavities under the ice shelves (Purich and England, 2021). However, CMIP6 models do not solve these cavities. Furthermore, mesoscale eddies, which are a source of ocean heat transport onto the shelves, are not well represented under the typical resolution used in CMIP6 models (Bracegirdle et al., 2020). This should improve with the new generation of models for CMIP7 using a higher resolution (Heuzé, 2021).

For the case of our research, we have used two models from CMIP6. CESM2-WACCM has strong warm biases, on and off the shelves (Purich and England, 2021). This model is also known to be affected by the representation of cloud feedbacks, which

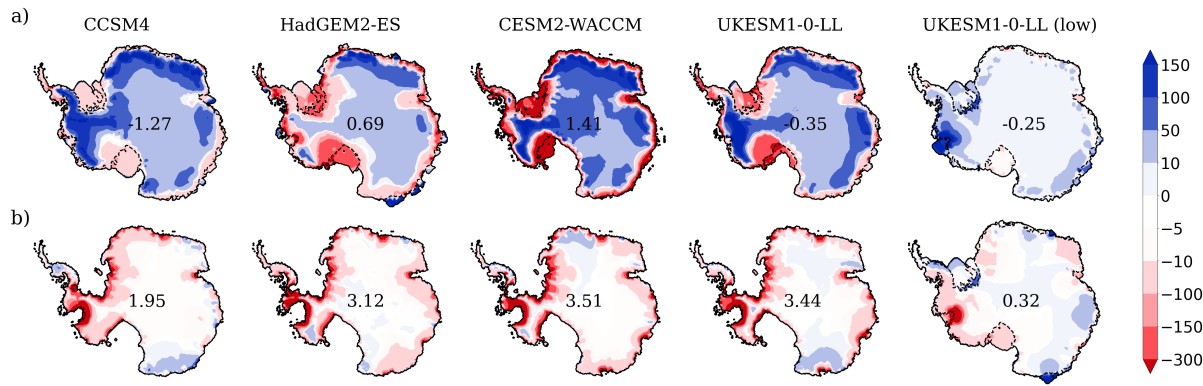

**Figure 11.** Ice-thickness anomalies (in m) for the atmosphere-only (a) and ocean-only (b) cases between the year 2500 and the start of the simulations. Solid lines and dashed lines represent the coastline and grounding line, respectively, at year 2500. Numbers included in the maps represent the total sea level contribution in meters for the medium value of the basal melting parameter.

are particularly strong yielding a larger climate sensitivity and therefore an inflated warming (Zhu et al., 2021). UKESM1-0-LL includes positive feedbacks over the Southern Ocean, inducing warming. In fact, for this model 40% of the global heat uptake occurs in the Southern Ocean and 28% of this uptake is transported northward (Williams et al., 2023).

     The separate effects of the atmosphere and the ocean on the AIS has recently been studied by Coulon et al. (2024) with comparable results to ours. The main driver of future AIS changes will be the ocean, whose interaction with the ice sheet also
happens to be the main source of uncertainty. In fact, only the ocean can contribute to sea level by more than 3 meters in 2500, affecting the WAIS predominantly. Meanwhile, in the short term, the atmosphere can help to somewhat balance the loss of ice mass through enhanced accumulation of ice. However in the long term it is expected that it will also contribute to the ice-mass loss through enhanced melting. Using the SICOPOLIS ice-sheet model, the medium value of the basal melting parameter and the original ISMIP6 forcing for 2100, Chambers et al. (2021) presented similar projections but keeping the climate constant
from 2100 to 3000, and Greve et al. (2022) proceeded extending the simulations from 2100 to 2300 with a climate index derived from simulations with the GCM model MIROC under RCP8.5 and RCP4.5 scenarios. Their results for the year 2300 and 2500 are very similar to ours with HadGEM2-ES and UKESM1-0-LL, but with the forcings from CESM2 and CCSM4, there is a clear mismatch (Table 4). Note that for the case of CESM2 in ISMIP6-2300, a different component is used for the atmosphere (WACCM) instead of that used for ISMIP6-2100 (CAM) that could explain this difference. This comparison also
makes evident that the procedure used to extend the simulations to the future is an important source of uncertainty.

     Lipscomb et al. (2021) considered a very similar procedure to ours (DIVA solver, basal friction law and optimization for thermal forcing, mainly). The main difference consists in the procedure used to extend the simulations until 2500. They use the ISMIP6-2100 dataset with a repeated forcing using the two-decadal range 2081-2100 in order to account for climate variability. Instead, we fix the forcing at 2300. As a consequence, their experiments, with the same MeanAnt calibration as ours, contribute
just over 400 mm SLE by year 2500, which is a much lower contribution than in our cases (more than 3000 mm SLE in two experiments). Even using the PIGL calibration, based on melt rates on Pine Island Glacier (Jourdain et al., 2020), and which

**Table 4.** Sea-level contribution (m SLE) at 2300 and 2500 (when data available) among the studies made by Chambers et al. (2021), Greve et al. (2022), Lipscomb et al. (2021) and the results from Yelmo with the medium value of $\gamma_0$.

| GCM Model/ Resolution | Chambers et al. (2021) 8 km | | Greve et al. (2022) 8 km | Lipscomb et al. (2021) 4 km | Yelmo 16 km | | Yelmo 32 km | |
|---|---|---|---|---|---|---|---|---|
| | 2300 | 2500 | 2300 | 2500 | 2300 | 2500 | 2300 | 2500 |
| CCSM4 | 0.5 | 2.1 | 1.25 | 0.28 | 0.1 | 0.5 | 0.16 | 0.6 |
| HadGEM2-ES | 1.6 | 3.6 | 3.3 | 0.35 | 1.5 | 3.2 | 1.6 | 3.5 |
| CESM2 | 0.45 | 1.75 | 0.95 | 0.26 | 1.7 | 3.5 | 1.9 | 3.8 |
| UKESM1-0-LL | 1 | 2.6 | 1.95 | 0.38 | 1.2 | 2.8 | 1.4 | 3.1 |

sets a $\gamma_0$ value one order of magnitude larger than the MeanAnt, they do not simulate a very high contribution, reaching roughly 1 m SLE by 2500. Therefore, our results indicate that the repeated forcing used in Lipscomb et al. (2021) leads to an underestimation of the AIS response. Berdahl et al. (2023) made a similar study to ours but not only varying $\gamma_0$ but also the
parameter p from Leguy's parameterization, which in our case is constant (p=0.5) in all experiments. In addition, except for CESM2, the GCMs they use as forcings are different from our case. For the experiment with CESM2, they found a sea-level rise of approximately 1600 mm SLE by 2500 while in our simulations, it reached more than 3000 mm SLE. This is explained by the same features pointed out for the results of Lipscomb et al. (2021) because they consider an average of the 2090-2100 period to generate the extension towards 2500. Table 5 shows the mean thermal forcing under the ice shelves (between 210
and 810 m of depth, see also Fig. A1) for the ISMIP6-2300 dataset in several periods used to extend the simulations in these studies and ours. Clearly the thermal forcing increases with time, so our setup allows for stronger thermal forcings that using the repeated forcing as in Lipscomb et al. (2021) or the averaged decadal period of Berdahl et al. (2023). For instance, for the UKESM1-0-LL case, the differences between our chosen period to extend the forcing and theirs can reach more than 4.5ºC. The same happens for CESM2-WACCM and HadGEM2-ES, reaching 3.5ºC approximately, and CCSM4 with roughly 2.3ºC.
Regarding the sensitivity study on $\gamma_0$ described in Berdahl et al. (2023), we reach similar conclusions. When increasing the value of $\gamma_0$, the retreat in the grounding line (especially in the WAIS) is triggered sooner. However, our initial ice-sheet configuration produces larger ice shelves compared to observations. The RMSE of the initial state with respect to observations, is about 200 m/yr and 200 m for ice surface velocity and ice thickness, respectively. Berdahl et al. (2023) obtained lower RMSE values, ~129 m/yr and ~58 m respectively (for the case they show with p=0). Nevertheless, our simulations were also
performed with a lower resolution than in these two works with the Community Ice Sheet Model (CISM, Lipscomb et al., 2019) using higher resolutions, but the sensitivity results should still be valid in this case, while the specific numbers could be more uncertain.

    We have also tested the same experiments with a $191 \times 191$ grid giving a 32-km resolution. Results for sea-level evolution are very similar, varying 7-10% for high-emission scenarios (Fig. A8). Generally, they yield an increase in the contribution

**Table 5.** Comparison of the mean thermal forcing (ºC) from the ISMIP6-2300 dataset on the ice shelves between depth layers of 210 and 810 meters depth, for the different time ranges used to extend the simulations (2081-2100, 2090-2100 an 2291-2300) in Lipscomb et al. (2021), Berdahl et al. (2023) and this work, respectively. Values relative to the reference given by Jourdain et al. (2020).

| | CCSM4 | HadGEM2-ES | CESM2-WACCM | UKESM1-0-LL | UKESM1-0-LL (low) |
|---|---|---|---|---|---|
| Lipscomb et al. (2021) \| 2081-2100 | 1.063 | 1.347 | 1.309 | 0.778 | 0.206 |
| Berdahl et al. (2023) \| 2090-2100 | 1.132 | 1.441 | 1.453 | 0.917 | 0.183 |
| This study \| 2291-2300 | 3.397 | 5.048 | 4.863 | 5.494 | 0.253 |

but a decrease in the spread between the low and high values of $\gamma_0$. Lipscomb et al. (2021) made the point with 2 and 4 km resolution with sea level variations of ∼15% and with 4 and 8 km with ∼20%, but they find that the higher the resolution, the higher the sea-level contribution. In our case, it is the contrary, at least at the resolutions we tested. In the same context, Sutter et al. (2023) argued using the Parallel Ice Sheet Model (PISM; Winkelmann et al., 2011) with 16, 8 and 4 km resolution grids that their 8-km ice-sheet configuration is close to their 16-km runs, nonetheless the rate of retreating at the grounding line is different, starting sooner for 8 and 4 km compared with 16 km, especially observed for Thwaites Glacier. This suggests the MISI instability is triggered earlier when using a higher resolution.

A critical point in our study is the initialization, which is based on the optimization during the first 15 kyr of the 20 kyr spin-up run of the thermal forcing correction ($\delta T$) and the bed friction coefficient (Section 2). As mentioned before, higher values of $\gamma_0$ lead to smaller ice shelves. As a consequence, lower values of $\gamma_0$ lead to higher positive values of the thermal forcing correction. To illustrate this phenomenon, the mean value over each drainage basin at the end of the spin-up is indicated in Figure A11a, showing a decrease with increasing $\gamma_0$. For instance, for $\gamma_0$=9620 m/yr the west of the Ronne-Filchner ice shelf illustrates the emergence of a positive anomaly, while for $\gamma_0$=14500 and $\gamma_0$=21000 m/yr, the thermal forcing correction is negative. Regarding the bed friction coefficient (Fig. A11b), it does not strongly depend on $\gamma_0$ on the EAIS. This is to be expected because the contact between the ice-sheet and ocean is limited in this area. However, the opposite occurs in the WAIS. For the highest $\gamma_0$ value, the basal friction coefficient over a wide part of WAIS (particularly around the divide) saturates at a constant value of 1, the imposed internal upper limit in Yelmo (lower limit is 0.001). This increases basal stress in that region, favouring shear-dominated flow and limiting the basal sliding.

Simulating the correct ice extent of ice shelves is a challenging task. Some ice models impose fixed ice-shelf extensions or prevent ice-shelves to grow beyond the observed ice front (Seroussi et al., 2019). In addition, tuning parameters to simulate realistic ice-shelf extensions may differ between different Antarctic embayments (Wilner et al., 2023). In this study, we simulate ice shelves prognostically, which leads to several ice-shelf fronts extended further than observations, particularly the Ronne-Filchner and the Ross ice shelves. For the Amundsen Sea Embayment, the ice shelf advances various kilometers into the ocean and that results in a deceleration of some ice streams flowing towards that sector at the beginning of the simulations (see Fig. 4 for a comparison with observations). Higher values of $\gamma_0$ results in a more realistic set up at the beginning (which can be

**Table 6.** Differences (in percentage) in sea level contribution at 2500 for the experiments carried out, between the runs with extended ice shelves and the ones with limited ice shelves.

| $\gamma_0(m/yr)$ | CCSM4 | HadGEM2-ES | CESM2-WACCM | UKESM1-0-LL | UKESM1-0-LL (low) |
|---|---|---|---|---|---|
| 9620 | 23.5 | -10 | 0.4 | 1.2 | 38 |
| 14500 | 13.5 | 4.3 | 1.9 | 4.7 | -48 |
| 21000 | 7.9 | 3.1 | 0.9 | 2.7 | -4.9 |

observed in terms of RMSE, Fig. 3), despite having a greater rate of melting. Since ice shelves are allowed to expand, we have done additional simulations considering limited ice shelves with the mask from observations in the BedMachine dataset. In terms of sea-level contribution, results and spreads are very similar for higher-emission scenarios (Fig. A9) with differences ranging between 0.4 and 23.5% in absolute value, but generally being less than 10% (Table 6). In the case of the low-emission simulation this difference is much greater. In general, the prognostic ice-shelf simulations have no considerable effect on the

simulated sea level, with a difference below 141 mm SLE at the end of the five-century period for the medium value of $\gamma_0$.

In relation to the two ISMIP6 protocols, we have also tested the goodness of fit of our simulations following ISMIP6-2300 with respect to the simulations made by Seroussi et al. (2020) for ISMIP6-2100, for the period 2015-2100 and the three values of $\gamma_0$ (Fig. A10). The spread and mean values produced by the different ice-sheet models ranges approximately between -80 and 260 mm SLE for different forcings and the medium value of $\gamma_0$ (Fig. 7 in Seroussi et al., 2020). Our results until the year

2100 fit between the shading produced for the values that we can compare for CMIP5, CCSM4 and HadGEM2-ES (Fig. A10), with both of them in the lower range of the spread. Our model results therefore show a similar low order of magnitude (always less than 100 mm SLE) in comparison to the sea-level contributions estimated in later centuries.

Throughout this work, the WAIS stands out as the main contributor to sea level rise in the future, with some sectors such as the largest ice-shelves, Ronne-Filchner and Ross, and Pine Island Glacier being particularly affected in some experiments

(Fig. 8). In these ice shelves, ice thickness is reduced more than 100 m in less than 100 years during the course of the 22nd century for high emission scenarios. Based on our results, we can say that the WAIS has great potential for large sea-level contributions in the future, independent of the associated uncertainty. The rapid WAIS ice loss is due to the debuttressing effect following ice-shelf thinning, which allows the interior ice flow to accelerate (Fig. A4), increasing the discharge. Nonetheless, it has recently been determined that snowfall, which is expected to increase in a warmer climate (Masson-Delmotte et al.,

2021b), also influences the imbalance in mass loss observed in this well-known region of Antarctica in the last decades, entailing variations in sea-level rise (Davison et al., 2023). The response of the solid Earth beneath Antarctica to the loss of ice, unevenly distributed on the WAIS and EAIS due to mantle properties, influences in long timescales the contribution to sea level as well, due to variations in grounding line locations and the bedrock (Whitehouse et al., 2019). In our simulations, we have used a simplified isostasy module considering constant mantle viscosities for the WAIS and EAIS, that could overestimate

and underestimate respectively for each region the contribution to sea level in 2500. This could be better assessed in the future

with a more realistic isostasy module that is currently being implemented (FastIsostasy, Swierczek-Jereczek et al., 2023; under review).

In our simulations for high-emission scenarios, the WAIS seems to start losing ice as soon as the beginning of the 22nd century with an increasing contribution to sea level rise until the end of the 23rd century, when this tendency is balanced with a lower rate despite the continuous loss of ice. Hill et al. (2023) applied a numerical stability analysis to show, using three different ice-sheets models, that under present conditions the AIS is not yet experiencing an irreversible MISI when activating and deactivating perturbations in basal melting. But reaching a certain point of retreat in the positions of the grounding lines, even with the current climate, could trigger an instability in a range of 300-500 years in some regions of the WAIS like the Amundsen Sea Embayment (Reese et al., 2022).

Further work is needed to better understand the structural uncertainty in future AIS mass loss projections. Our study has just focused on the parametric uncertainty that arises within one basal melt parameterization, the quadratic non-local basal melting law of Jourdain et al. (2020). Other basal melting parameterizations include the Potsdam Ice-shelf Cavity mOdel (PICO; Reese et al., 2018). In Burgard et al. (2022), most existing parameterizations have been assessed considering the ice-shelf slope relative to the horizontal too. Also it could be tried another procedure to guess the rates of basal melting and it has been proved in recent times that machine and deep learning have come in handy in that matter (Rosier et al., 2023). Finally, it is important to note that our offline approach can affect projections of the AIS contribution to sea level because of a misrepresentation of the feedbacks between warm water intrusions, basal melting and other coupling processes (Park et al., 2023). In fact, as shown by Golledge et al. (2019), the freshwater fluxes from AIS melting could confine warm water in the subsurface of the Antarctic Ocean leading to higher basal melting, therefore affecting sea-level projections in coupled ocean-ice-sheet models. This meltwater impacts the ocean's overturning circulation, precipitation patterns and temperature variability across the world (Bracegirdle et al., 2020). Eventually these opposing effects should be addressed by the use of coupled ocean-ice models.

## 5    Conclusions

The effect of the uncertainty in the climate forcing in four high-end scenarios and one low-end scenario on the response of AIS to future climate change has been explored in this work. First, we have shown that oceanic forcing is the main driver of mass loss in the WAIS and thus the AIS overall. Thus, simulations driven by similar atmospheric forcing but different oceanic conditions can produce very different sea-level contributions for the future. This is illustrated by the UKESM1-0-LL and CCSM4 GCM models that under a high-emission scenario are associated with contributions of almost 2.7 m and 0.5 m respectively by 2500 due to the very different oceanic forcing. In fact, we demonstrated that a low emission scenario could be comparable to the case of CCSM4. Therefore, at least on short timescales (hundreds of years), a correct representation of the ocean response to future climate change is critical.

We also analyzed the parametric uncertainty arising from the basal melt law used to force our ice-sheet model, the heat exchange velocity $\gamma_0$ between ice and ocean on the ice shelves. Our results show that uncertainty in Antarctica projections is profoundly linked to the selection of this parameter and to the effect of the ocean rather than the atmosphere. Differences

between choosing a low and a high value could alter sea level more than 2 m in 2500 in some experiments. Nevertheless, this contribution to sea level is unevenly distributed spatially along the ice sheet. The west region is robustly projected to generate a positive contribution to the rise of sea level, due to its marine ice-sheet condition and the physical effect of the buttressing of the main ice shelves. Meanwhile accumulation zones are identified in the stable east part, which gain ice mass, but most notably in some cases even this region has a positive contribution to sea-level rise contributing to the sea level through the weakening of some ice shelves.

Regarding the WAIS, this region experiences a continuous loss of ice starting roughly in the middle of the 22nd century and extending beyond 2300 when most of the ice disappeared including the main ice shelves, Ross and Ronne-Filchner. Furthermore, the main reason for these changes can be attributed directly to the effect that the ocean has over the AIS.

*Code availability.* The Yelmo ice sheet model code is additionally available at https://github.com/palma-ice/yelmo (Robinson et al., 2020)

*Data availability.* The netCDF files containing all the results from simulations together with a sample of the namelist of parameters used are available at this Zenodo repository: 10.5281/zenodo.10657938

## Appendix A: Additional figures

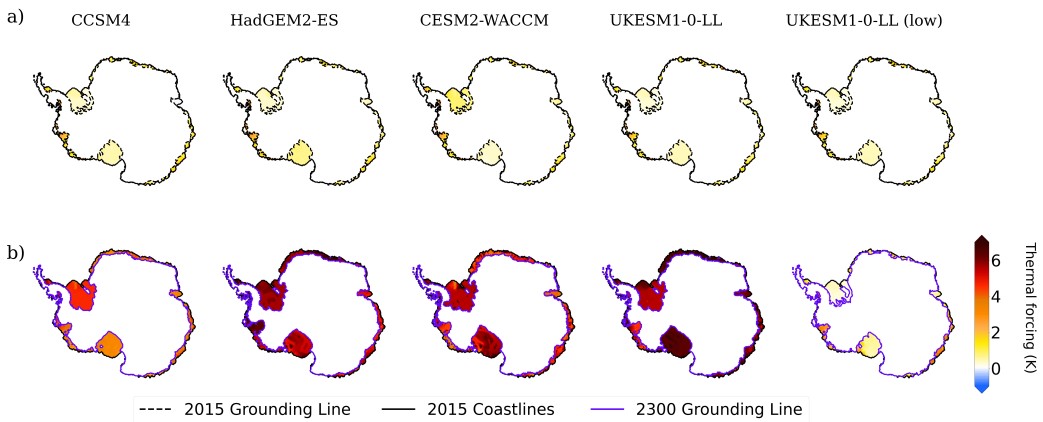

**Figure A1.** Depth-averaged thermal forcing under the ice shelves (K) between 210 and 810 meters in (a) 2015 and in (b) 2300.

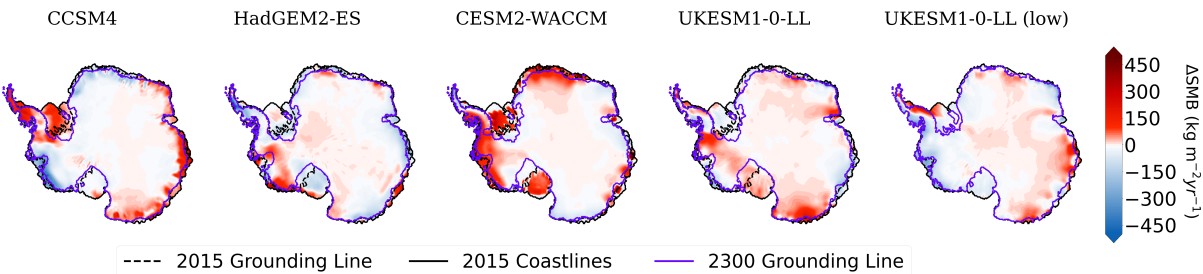

**Figure A2.** SMB (kg m$^{-2}$ yr$^{-1}$) anomaly for the different GCM models between 2300 and 2015.

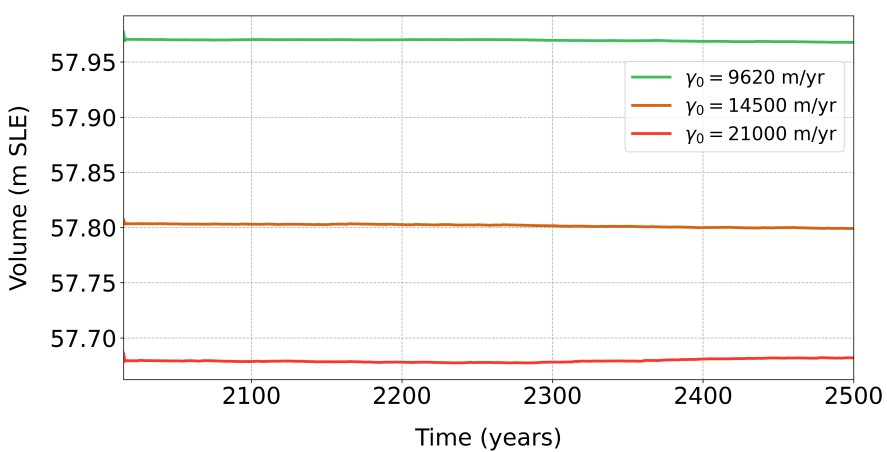

**Figure A3.** Evolution of the ice volume above flotation (m SLE) in the control runs for the AIS, with the different values of $\gamma_0$.

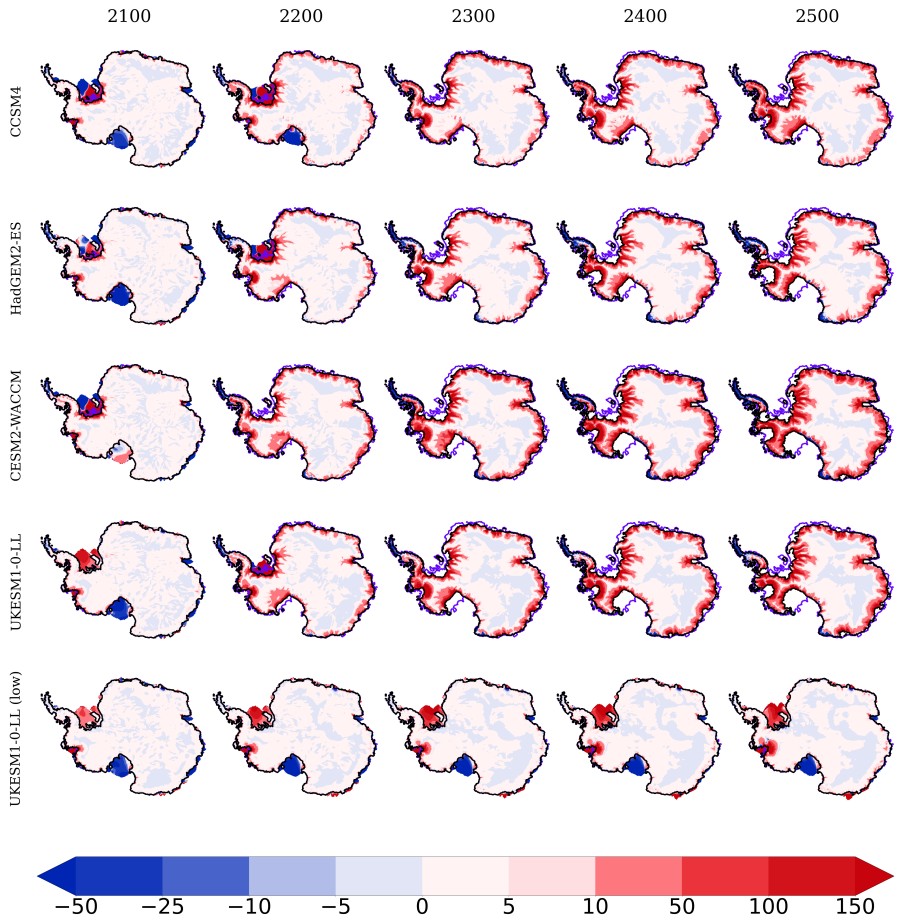

**Figure A4.** As Figure 8 but for ice surface velocity anomalies (in m/yr). Note that only the cells where there is ice (floating and grounded) at the specific times are represented.

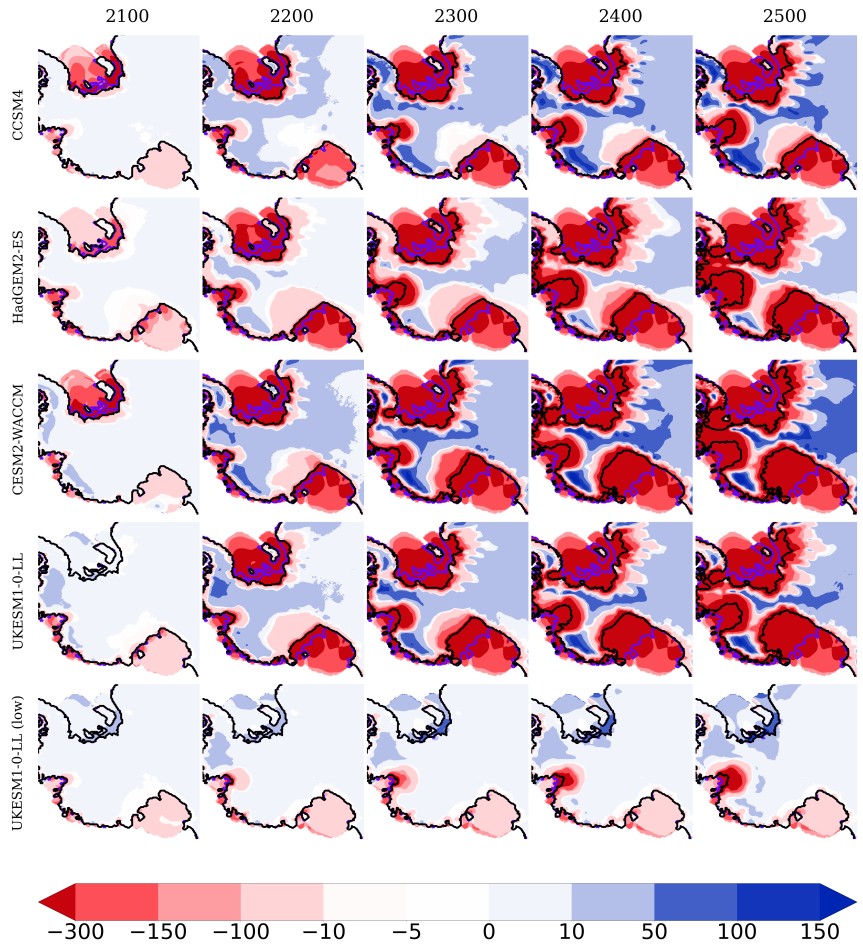

**Figure A5.** Ice thickness anomalies (in m) in the WAIS (defined as in Fig. 4a) for the years 2100, 2200, 2300, 2400 and 2500 with respect to the start of the simulations for the set of five experiments carried out with the medium value of $\gamma_0$. The grounding line is represented by black and violet colours for 2500 and 2015, respectively. To show negative anomalies in the WAIS, no masking has been implemented as time evolves, keeping the original coastlines.

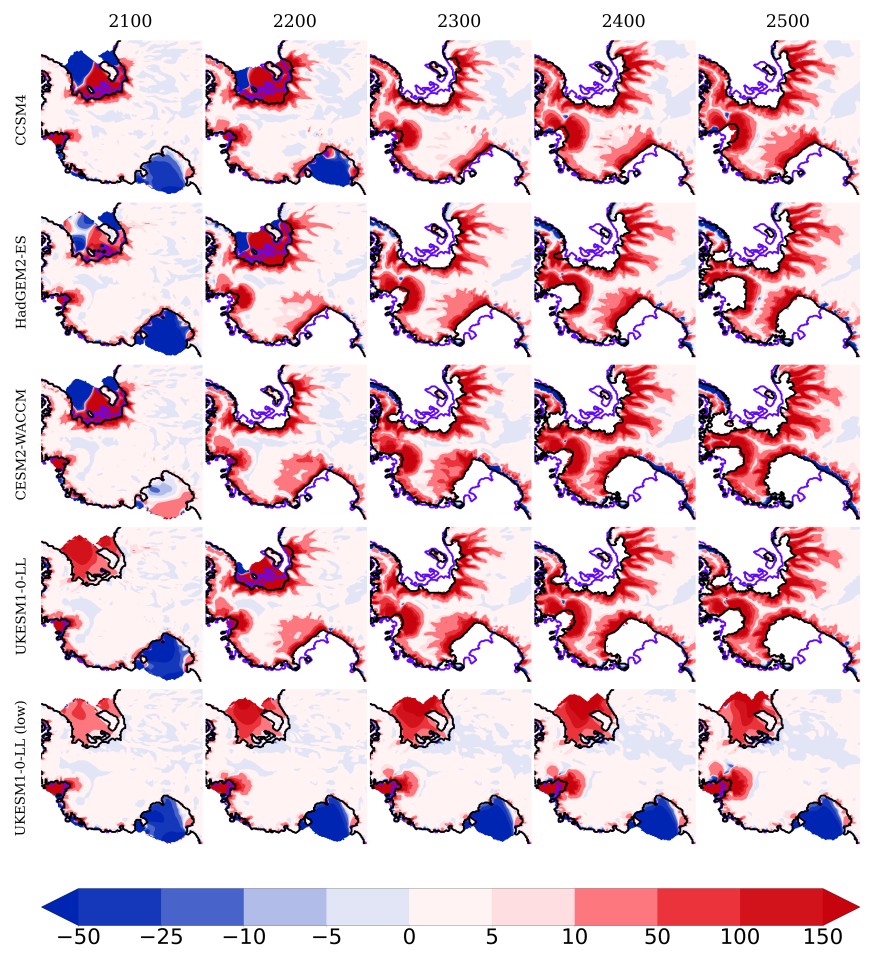

**Figure A6.** As Figure A5 but for ice surface velocity anomalies (in m/yr).

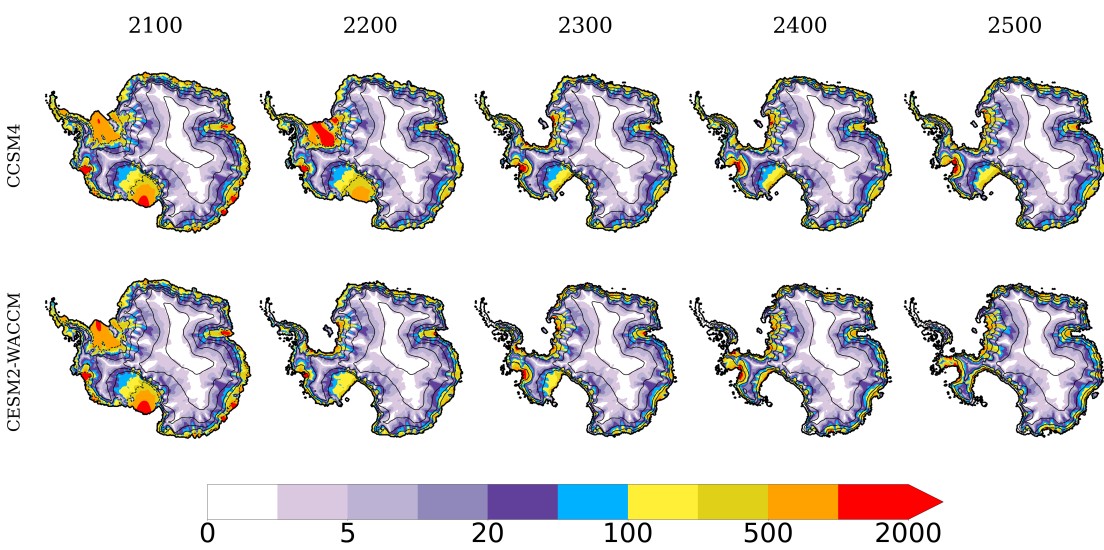

**Figure A7.** Ice surface velocity evolution (in m/yr) at the beginning of each century for the experiments with the GCM models CCSM4 (top) and CESM2-WACCM (bottom). Solid lines represent the coastline and surface elevation levels at the year considered while dashed lines delineate grounding lines.

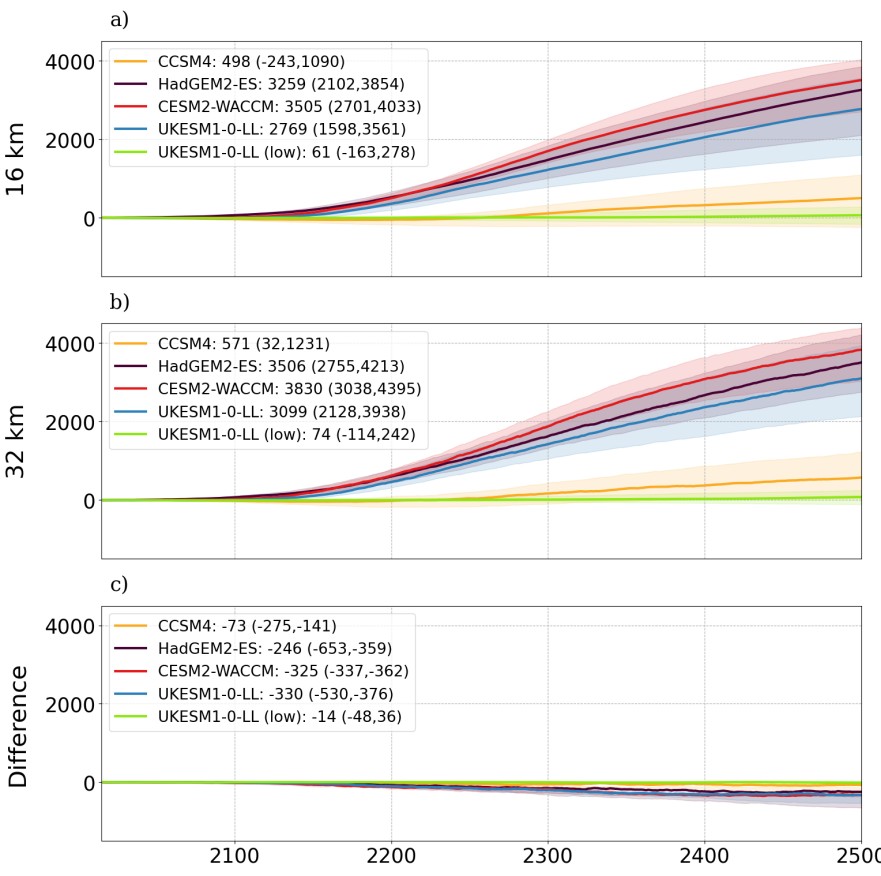

**Figure A8.** Sea level contribution for the different experiments carried out and spread of the values of $\gamma_0$ using resolutions of 16 km (a) and 32 km (b). In c, differences between a and b.

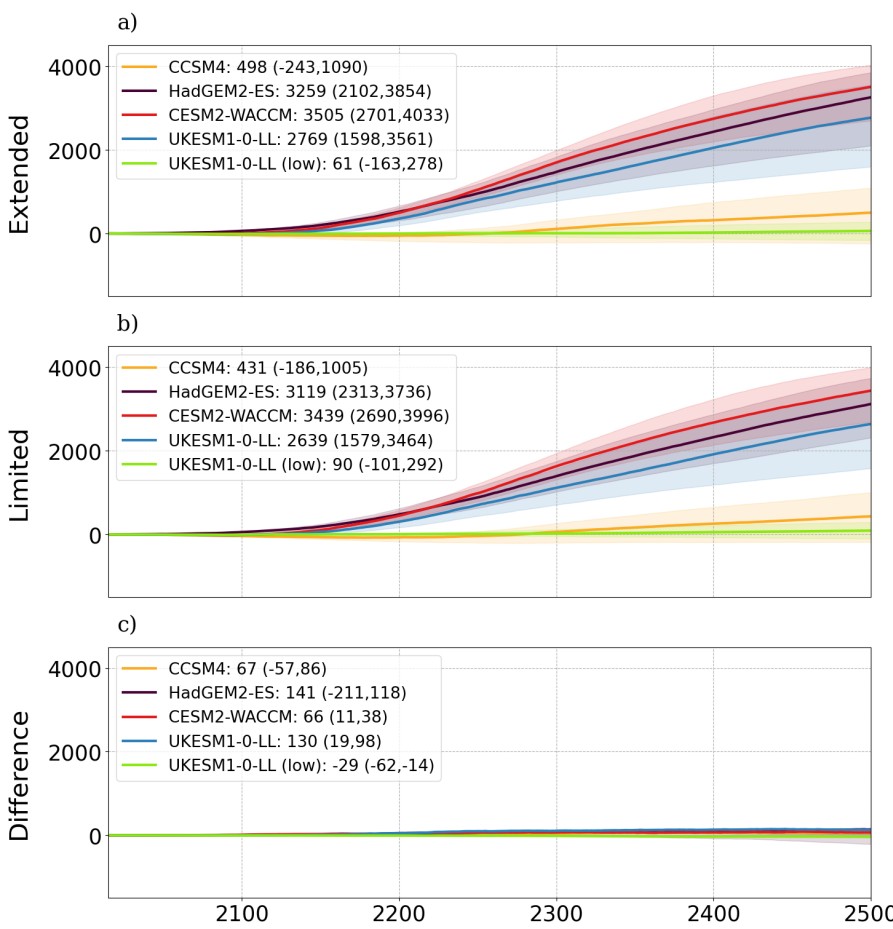

**Figure A9.** Evolution of the sea level contribution until 2500 for the different experiments carried out, together with the spread produced by the different values of $\gamma_0$. In a with the extended ice shelves, in b with the ice shelves limited by a mask from observations in the BedMachine dataset, and in c difference between a and b.

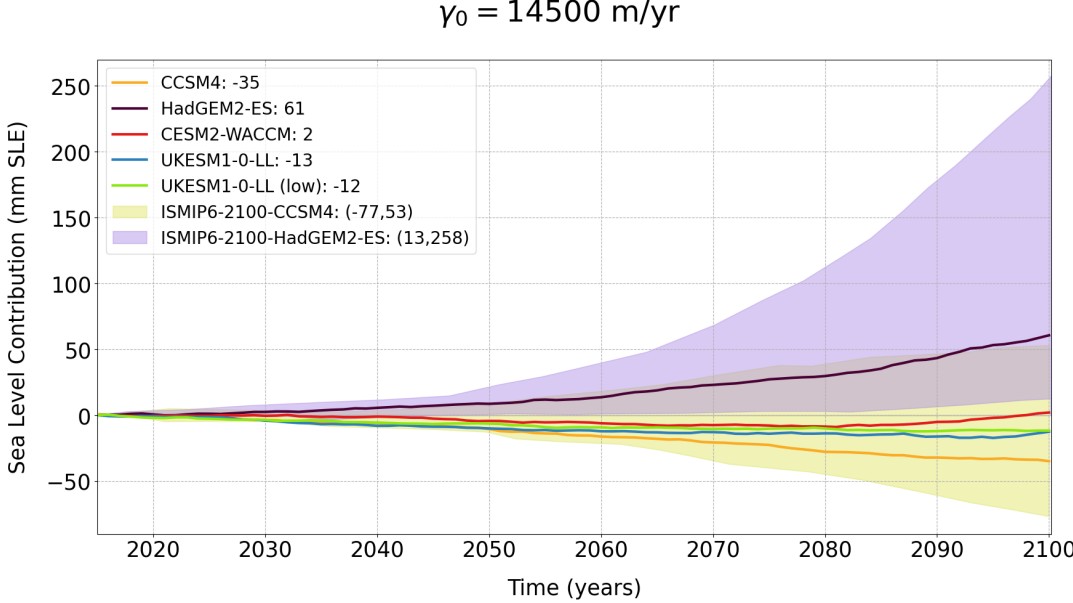

**Figure A10.** Sea level contribution (mm SLE) for the experiments carried out with five GCM forcings under high-emission scenarios, until the year 2100 for the medium value of $\gamma_0$. Solid lines represent the curve for the values of $\gamma_0$ and the shading the overall results from Seroussi et al. (2020) with different ice-sheet models for the forcings given by CCSM4 and HadGEM2-ES.

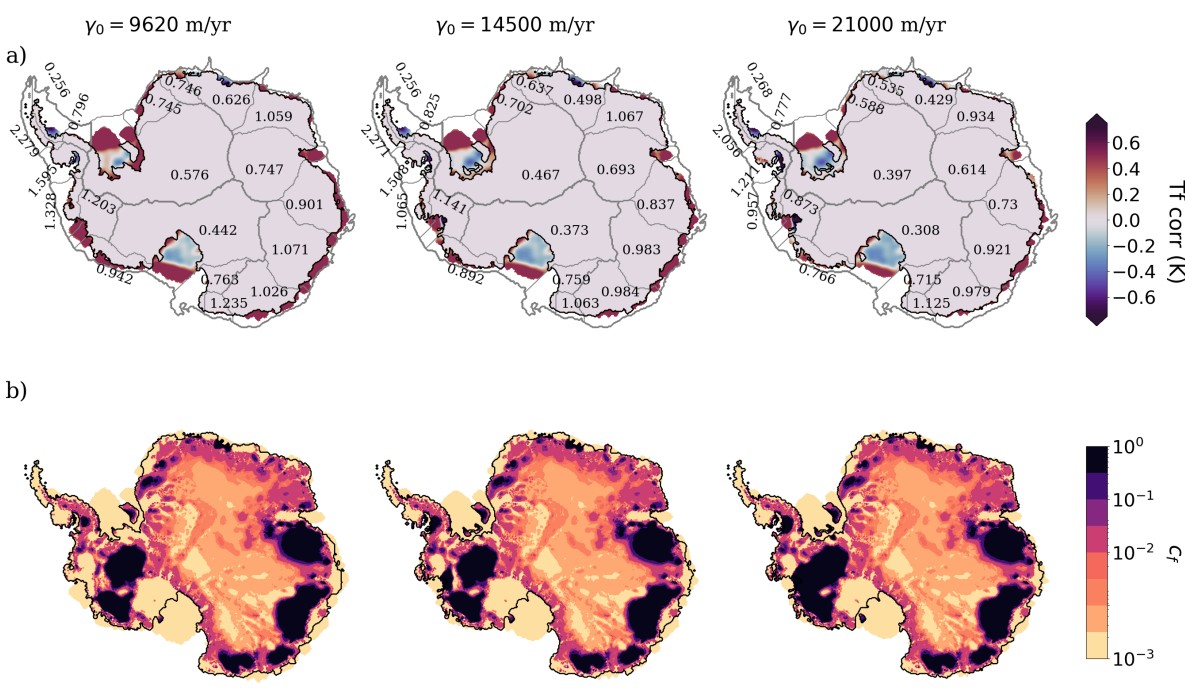

**Figure A11.** a) Thermal forcing correction (K) at the beginning of the control simulation for the three values of $\gamma_0$. In grey, basins in which Antarctica is divided and numerical values with the average thermal forcing on each basin at start. b) Bed friction coefficient at the beginning of the simulations. The minimum value is 0.001 set as a limiting parameter in Yelmo.

*Author contributions.* A. J-M. carried out the simulations. A. J-M., J. B. and A. R. implemented the ISMIP6 protocol for driving the Yelmo ice-sheet model. All co-authors analyzed the results. A. J-M. prepared the manuscript with contributions and revisions from all co-authors.

*Competing interests.* At least one of the (co-)authors is a member of the editorial board of The Cryosphere.

505 *Acknowledgements.* This research has been supported by the Spanish Ministry of Science and Innovation (project MARINE, grant no. PID2020-117768RB-I00) and by the European Union Horizon 2020 research and innovation programme (grant no. 820970, TiPES). Antonio Juarez-Martinez was funded from the predoctoral grant partnership between the Complutense University of Madrid and Banco Santander (CT58/21-CT59/21). Alexander Robinson received funding from the European Union (ERC, FORCLIMA, 101044247). All simulations were performed in Brigit, the HPC server of the Faculty of Physics of the UCM. We also acknowledge the projects CliC, CMIP5/CMIP6 and

510 ISMIP as being the main sources for our work, and the two reviewers that have contributed with valuable suggestions and revisions to create the final version of this manuscript.

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
