# Peer review of "Antarctic sensitivity to oceanic melting parameterizations"

_EGUsphere, 2023_

## Referee Comment (RC2)

Juarez-Maritnez et al. investigated the impacts of emission scenarios, the climate forcings and the heat exchange velocity in the basal melt parameterisation on the projected Antarctic mass loss before 2500 They found that the Antarctic ice sheet contributions to sea level rise highly depend on the heat exchange velocity at the ice-ocean interface and also the climate forcing chosen at high emission scenario. Separate atmospheric-only and oceanic-only experiments show that oceanic forcing plays a dominant role for the West Antarctic while atmospheric forcing is more important for the eastern sector and the interior. Overall, the manuscript is well written. However, the model setup section and some of the descriptions on results need improvements. I'm happy to support the publication of this study after major revisions.

Here are some general comments:

In the model description, some information is missing, including how the ice front position is updated, how the basal drag is treated for partially-floating cells, the reference SMB dataset. About the moving ice front positions, how the ice front position is updated is not described at all. Is a calving included? Could you please show the difference between the ice front position after spin-up and the present-day? How would the difference affect your results?

Coarse resolution like 16 km would largely affect the estimation of mass loss and also the movement of grounding line, which would influence the basal drag and basal melting applied at the grounding line. However, nothing related with this has been discussed, which I think is important.

The section 3.1 needs to be restructured. You discuss the sensitivity to $\gamma_0$ in 1st paragraph and then discuss the sensitivity to different forcings under a medium value of $\gamma_0$ for the rest of the section. However, about the sensitivity to different forcings, you jump between different regions, different GCM forcings, low and high emissions, which is quite chaotic. The focus of each paragraph is not clear to me. About the summary paragraph in Line 245-246, it's a repeat of caption rather than a summary. I think Fig 9 is a good plot which can help you explain things earlier, which should be combined in your result description earlier.

The Sec. 3.3 describes the individual effect of the atmosphere and the ocean on the projected ice mass loss from west and east Antarctica. However, it is not enough evidence to say that atmosphere is the dominant factor to margin of East Antarctic Ice Sheet, especially for Cook Ice shelf region, Totten Glacier region and Amery Ice Shelf region. These regions are losing ice clearly and show rounding line retreat in Fig 13b. Although it is not as obvious as west antarctica, it should be discussed at least.

There are 13 figures and 4 tables in this manuscript, some of the figures are not very crucial to be included in the main manuscript, which can be moved to the supplementary material.

The resolution of some figures (Fig. 7, 8, 9, 11) is pretty low and needs to be improved. Some of the text labels on the figures appears.  due to the low resolution of figures.

Specific Comments:

L21-22: citation please

L28: what is 'ejecting circa'?

L45-46: need to add citation Edwards et al., 2021

L50: ice sheet stability?

L68: ISMIP6 is short for The Ice Sheet Model Intercomparison Project for CMIP6

L115: citation for this regularized Coulomb sliding law please.

L130: you should explain specifically what $\gamma_0$ is here.

L163: is → are

L169: which GCM forcing is chosen here for period 1995-2014?

L181: (Lipscomb et al., 2021) → Lipscomb et al., (2021)

L190: imposed → kept constant?

L198-199: refer to Fig. 4a here.

L205: ocean forcing from CESM2 is not the maximum here compared with UKESM and HadGEM2

L208: 'the low-emission case' with UKESM

L213: suggest 'we can see the sea level contributions slows down in the last two centuries.'

L215: I think they are all nearly above 5 mm SLE/yr while HadGEM2 is lose to 10 mm SLET/yr.

L219: for the medium value of $\gamma_0$, no experiments reach 3 m at 2500.

L221-222: 75% + 20% is not 100%.

L227: Fig 1 did not show anything related with Amundsen and Bellingshausen sea sectors.

L228: for Amery, you need to cite Fig 7 rather than Fig 6.

L230: why do you only mention Ronne-Filchner here? Amery also shows loss of ice only for CSEM2 and CCSM4.

L233: it is not obvious to me in 2200. 2300 is more obvious.

L235: not for CCSM4

L236: what do you mean "reaching well above 300m"? the thickness?

L237: why do you exclude the PIG and Thwaites here?

L240: what do you mean "Ronne-Filchner ice shelf is larger"? If you are talking about readvance of GL (which I can't tell it well from current resolution of figure) in these regions, the ice shelf area is smaller.

L242: The ice flow is decelerating in ROSS under low emission scenario.

L253: this confused me a lot. If the ice shelf disappears from 2300 onward, why we still see floating regions in Figure 7 after 2300. From Figure7 ,the ice shelf is always there.

L263: I can not tell mass loss from WAIS changes its tendency from Fig 11.

L279: When you say 'The EAIS ice shelves also have their ice mass reduced', which plot are you talking about? Or you mean the margin of EAIS?

L280: 'in the interior and eastern areas'→'in the eastern interior'?

L297: the forcing is stopped? I thought it was just kept constant.

L312: in more than 3 meters → by more than 3 meters; 'affecting predominantly to the WAIS' → 'affecting the WAIS predominantly'

L314: I don't understand this sentence. Do you mean through enhanced surface melting?

L319: There is also a clear mismatch between your result and Greve et al. 2022 in Table 4.

L321: Then what about the mismatch in CCSM4?

L346: unfished sentence.

L366: on Eq (3). Or you don't need to mention it here.

L372: I think you mean a position contribution to SLR. But by saying 'negative effect' can be confusing.

Figures

Figure 2: Caption: the differences in thermal forcing for b).

Figure 7 & 8: I did not see the coastlines at all, which is a very import feature to be shown clearly in the figure.

Figure 8: what do you mean the cells where there is ice? I think you mean grounded ice? If yes, why the floating ice is included in UKESM LOW emission case?

Figure 12: in the AIS → from the AIS

References

Edwards, T.L., Nowicki, S., Marzeion, B. et al. Projected land ice contributions to twenty-first-century sea level rise. Nature 593, 74–82 (2021). https://doi.org/10.1038/s41586-021-03302-y

---

## Author Comment (AC1)

**Review 1 of Antarctic sensitivity to oceanic melting parameterizations**

This paper uses the recently produced extended ISMIP6 Antarctica oceanic and atmospheric forcings which now extend to year 2300 (as opposed to year 2100 previously) and investigate the impact of these forcing on Antarctica using the YELMO ice sheet model. In their study, the authors not only consider the impact of the model when forced with both the oceanic and atmospheric forcings but also when forced by only either of them. They concluded that oceanic forcings greatly affect the West Antarctic Ice Sheet (WAIS) and not so much the East Antarctic Ice Sheet (EAIS) and vice versa for the atmospheric forcing.

This paper gives more insights on the capability of the YELMO ice sheet model in performing future projection experiments. While analyses on the impact on the ice sheet in separating the oceanic and atmospheric forcing from the ISMIP6 is not well published, previous work done on oceanic forcing impact on the AIS that could have been discussed in greater length in this manuscript.

In addition, the discussion of the initialization strategy and its impact on the simulation deserve to be expanded on many levels from its details and result demonstrations to the impact on the forcing interpretations.

Very similar work of this nature has been done before by Lipscomb et al. (2021) and Berdahl et al. (2023) and the authors did not contrast or conform their work with theirs (I will be specific on this matter in my comments afterwards). Also, I believe the authors missed a few opportunities in this manuscript: they could have added more analyzes focusing more on the impact of heat at the grounding line compared to under the ice shelves (which they hinted at in Fig1 but never really pinpointed thereafter); they could have compared their results to model running at higher resolution and making the point that a 16km-resolution model performs relatively well in these types of experiments compared to model using higher resolutions. Regarding the latter point, they could have analyzed the impact of the first 100 year forcing on their model and compare with the results from Seroussi et al. (2020) since they did not participate in this community experiments.

Finally, this manuscript uses forcings that were created for a community effort, the ISMIP6 AIS-2300 which is an extension of the original ISMIP6 that only provided forcing and simulations until year 2100. To the best of my knowledge, the extended dataset is not yet publicly available and should be used with restrain. While the wiki page for the extended experiment is available for now, it only describes the extended protocol but does not provide a link to the data. I do not see reference credited for these forcing. The manuscript of the extended community experiments was not submitted at the time of this article's submission and was, in fact, just submitted recently. In the absence of citation, and as a courtesy and recognition of the work into generating the dataset, the authors should reach out to the core ISMIP6 organizing group on how to best cite or acknowledge the extended dataset. Please, be mindful about these types of details in the future. For now, please refer to Nowicki et al. (2020), Jourdain et al. (2020), Barthel et al. (2020) (since these papers show the forcing to 2100 and the same scripts were

used to prepare the data) and Seroussi et al. (2024) (under review). For further information on ISMIP6 AIS dataset, I would refer the co-authors to the ghub page: https://theghub.org/groups/ismip6/wiki/MainPage/ISMIP6Projections2300Antarctica and the section "how to obtain the 2300 AIS ISMIP6 dataset" for more information.

I would support the publication of this manuscript after major revisions.

Thank you for your comments and feedback. We will ensure that the new version of the manuscript takes into account all of your comments and suggestions. In particular, we have contrasted our work in larger depth with that of Berdahl et al (2023), we have deepened our analysis of the impact of basal melting at the grounding line and we have reached out to the ISMIP6 organisers to properly manage citation of the community dataset, which was indeed a grave oversight on our side. They have told us that the best way to proceed is citing Nowicki et al., 2020, Jourdain et al. 2020 and Seroussi et al. 2020, together with the new submitted paper for the 2300 forcings and results, Seroussi et al. sub. They also commented that we have to be aware that in the next months the status of this last paper could change and in that case it would be necessary to change the citation if published by that time (Seroussi, pers. comm.). In addition, they have pointed out to acknowledge the projects CliC, CMIP5/CMIP6 and ISMIP, and we have done so.

In the text below, we respond to your specific points and we provide new figures for the comparisons you mention and redo the current figures for the preprint. Note that our comments are in blue.

**Specific comments**

As mentioned above, a lot of this work feels like an extension of Lipscomb et al. (2021) and Berdahl et al. (2023) who carried out similar experimental setup and parameter sensitivity studies using a very similar initialization procedure. Using the original ISMIP6 oceanic forcing only dataset Lipscomb et al. (2021) not only showed the accelerated retreat behavior before year 2100 also shown in this study, they also show very similar patterns of retreat. Since they are using all the ocean melt calibrations from ISMIP6, please discuss your results with respect to theirs as they also ran to year 2500, like you. These additions can be done in the discussion sections and in the introduction (lines 75-84).

In the new version of the manuscript we will include a detailed comparison with the studies of Lipscomb et al. (2021) and Berdahl et al. (2023) in the introduction and discussion sections focusing on the similarities but also the differences in our experimental setup.

Specifically, Lipscomb et al. (2021) considered a very similar procedure to ours (DIVA solver, basal friction law and optimization for thermal forcing, mainly). The main difference consists in the procedure used to extend the simulations until 2500: they proceed in a similar way as Berdhal et al. (2023), but using the ISMIP6-2100 dataset and with a repeated forcing using the two-decadal range 2081-2100 in order to account for climate variability. Instead, we fixed the forcing at 2300. As a consequence,

their experiments, with the same MeanAnt calibration as ours, contribute just over 400 mm SLE by year 2500, which is a much lower contribution than in our cases (more than 3000 mm SLE in two experiments). Even using the PIGL calibration, which sets a $\gamma_0$ value one order of magnitude larger than the MeanAnt, they do not simulate a very high contribution, reaching roughly 1 m SLE by 2500. Therefore, our results indicate that the repeated forcing used in Lipscomb et al. (2021) leads to an underestimation of the AIS response. Table A1 shows the mean thermal forcing under the ice shelves (between 210 and 810 m of depth) for the ISMIP6-2300 dataset in several periods. Clearly the thermal forcing increases with time, so our setup allows for stronger thermal forcings that using the repeated forcing as in Lipscomb et al. (2021). For instance, for the UKESM1-0-LL case, the differences between our chosen period to extend the forcing and theirs can reach more than 4.5ºC. The same happens for CESM2-WACCM and HadGEM2-ES, reaching more than 3.5ºC, and CCSM4 with roughly 2.3ºC.

Although our final  amplitudes of the sea-level response differ, both studies agree in showing an initial slow increment (see Figure A7 on a comment below) and produce a significant loss of ice by the end of the 21st century.

For the new version of the article, we will incorporate this comparison in the discussion section, including Table A1.

| Reference \| Time range/ GCM Model | CCSM4 | HadGEM2-ES | CESM2-WACCM | UKESM1-0-LL | UKESM1-0-LL (low) |
|---|---|---|---|---|---|
| Lipscomb et al., (2021) \| 2081-2100 | 1.063 | 1.347 | 1.309 | 0.778 | 0.206 |
| Berdhal et al. (2023) \| 2090-2100 | 1.132 | 1.441 | 1.453 | 0.917 | 0.183 |
| This study \| 2291-2300 | 3.397 | 5.048 | 4.863 | 5.494 | 0.253 |

Table A1. Comparison of the mean thermal forcing  (ºC) from the ISMIP6-2300 dataset on the ice shelves between depth layers of 210 and 810 metres depth, for the different time ranges used to extend the simulations (2081-2100, 2090-2100 an 2291-2300) in Lipscomb et al. (2021), Berdhal et al. (2023) and this work, respectively. Values relative to the reference given by Jourdain et al. (2020).

Berdahl et al. (2023) extended Lipscomb et al. (2021) by doing a sensitivity study on thermal forcing (TF) and basal friction parameters. The thermal forcing parameter they varied in their study was \gamma0, just like you and they concluded that not only ocean thermal forcing played a more important role than basal friction law in future forcing simulations, but that increasing gamm0 led to more melting and grounding-line retreat, just like what you find. Please discuss your finding in comparison to theirs.

We have included a comparison of our results with Berdahl et al (2023) in the new version. They made a similar study to ours but not only varying $\gamma_0$ but also the parameter p from Leguy's parametrization, which in our case is constant (p=0.5) in all experiments. In addition, except for CESM2, the GCMs they use as forcings are different from our case. For the experiment with CESM2, they found a sea-level rise of approximately 1600 mm SLE by 2500 while in our simulations, it reached more than 3000 mm SLE. This is explained by the same features pointed out above in the comparison against Lipscomb et al. 2021: while we extended our simulations to 2500 by fixing the forcing to 2300, they considered an average of the 2090-2100 period. This could lead to an under-estimation compared to our case. Again, in Table A1, the differences for the ISMIP6-2300 dataset on the ice shelves are shown to compare their mean values in their period (2090-2100) with ours. As a result, as in Lipscomb et al. (2021), their thermal forcing is more than 3 ºC below ours for some experiments.

Regarding the sensitivity to $\gamma_0$, we reach similar conclusions. When increasing the value of $\gamma_0$, the retreat in the grounding line (especially in the WAIS) is triggered sooner. However, as we will discuss later, our initial ice-sheet configuration produces larger ice shelves compared to observations. The RMSE of the initial state with respect to observations, is about 200 m/yr and 200 m for ice surface velocity and ice thickness, respectively. Berdhal et al. (2023) obtained lower RMSE values, ~129 m/yr and ~58 m respectively ( for the case they show with p=0). Nevertheless, our simulations were also performed with a lower resolution (16 km) than in their work with CISM (4 km). We believe the sensitivity results should still be valid in this case, while the specific numbers could be more uncertain.

In the revised text, we will add a comparison with Berdhal et al. (2023) for the discussion part of the text and incorporate Table A1 as the main source for contrasting the differences that could arise between their work and ours.

Your discussion on the initialization deserves additional details:

Thank you for pointing this out. We will add the following details and a more complete description to the revised text:

1. What do you use for calving?

    We have used the von Mises calving law (Lipscomb et al. 2019; Morlighem et al 2016) with the scaling calving parameter $k_t$=0.0025 m/(yr Pa) and $w_2$=25 for the calving eigenvalue weighting coefficient, as in the approach by Lipscomb et al. (2019). The calving limit in ice thickness is set to 200 m.

2. What data are you using for Surface Mass Balance (SMB) and air temperature?

    For Surface Mass Balance (SMB) and 2m-air temperature we have used monthly values from the Regional Atmospheric Climate Model (RACMO v2.3) together forced by ERA-Interim reanalysis mean climatology dataset from 1981-2010. We will include this in the updated manuscript version.

3. What do you use for your geothermal forcing?

We use Shapiro and Ritzwoller (2004), just as Berdhal et al. (2023) with CISM. We will include this in the updated manuscript version.

4. What data are you using for comparing your ice velocity results?

For ice thickness observations, we use Morlighem et al. (2020) and for ice surface velocities, Rignot et al. (2011). We will include this in the updated manuscript version.

5. In many places you are referring to Bedmachine. If the climatic and ice velocity dataset you are using are similar than the ones used in Bedmachine, please cite them separately as they were not produced by Bedmachine!

Yes, we forgot to refer to Rignot et al. (2011) for ice surface velocities and will make sure to do so. We will include this in the updated manuscript version.

6. How does varying \gamma0 influence the inverted parameters?

Figures A1 and A2 show the thermal forcing and bed friction coefficient fields that are obtained from the optimization process. All $\gamma_0$ values produce extended ice shelves at the start. The higher the value of $\gamma_0$ is, the smaller the ice shelves, and therefore the closer to observations in terms of the extended area and RMSE (lower values). The reason for this is that we do not artificially limit the extent of our ice sheets as other groups do. This yields large ice shelves as compared to observations and to other groups.

As a consequence, lower values of $\gamma_0$ lead to higher positive values of thermal forcing correction. To illustrate this issue the mean value over each drainage basin is indicated, showing a decrease with increasing $\gamma_0$. For instance, for $\gamma_0$=9620 m/yr the west of the shelf illustrates the emergence of a positive anomaly, while for $\gamma_0$=14500 and $\gamma_0$=21000 m/yr, the thermal forcing correction is negative (Fig. A1).

The optimised bed friction coefficient (Fig. A2) does not strongly depend on $\gamma_0$ in the EAIS. This is to be expected because the contact between the ice-sheet and ocean is limited in this area. However, the opposite occurs in the WAIS. For the highest $\gamma_0$ value, the basal friction coefficient over a wide part of WAIS (particularly around the divide) saturates at a constant value of 1, the imposed internal upper limit in Yelmo (lower limit is 0.001). This increases basal stress in that region, favouring shear-dominated flow and limiting the basal sliding

We will include this discussion in the updated manuscript version.

[Figure]

Figure A1. Thermal forcing correction (K) at the beginning of the control simulation for the three values of $\gamma_0$. In grey, basins in which Antarctica is divided and numerical values with the average thermal forcing on each basin at start.

[Figure]

Figure A2. Bed friction coefficient (parameter optimised during the spin-up) at the beginning of the simulations. The minimum value is 0.001 set as a limiting parameter in Yelmo.

7. You mention that your spin-up last 20 kyrs, during which the first 15 kyrs are used to invert for observation. What do you do for the remaining 5kyrs?

During the first 15 kyr, the basal friction and thermal forcing fields are optimized. During the last 5 kyr, the optimised fields are held constant and the spin-up is run with the reference climatology towards steady state. We will make this point clear in the revised text.

Please show a map of \deltaT_sector resulting from your inversion process. This map can contain the values for all 3 \gamma0. Please discuss in more length the implication of this coefficient. In particular, if you need to cool down your basin by several degrees to obtain a stable ice sheet state, it means (at least one

interpretation could be) that the thermal forcing used in the inversion are too warm to obtain a steady state with current observations and the melting of ice shelves is underway.

Figure A1 shows the thermal forcing correction over the shelves with its mean average value over the different sectors of the AIS indicated, just as in Fig. 5 from Jourdain et al. (2020), at the beginning of the simulations (end of the spin-up process), for the three different values of $\gamma_0$ used. The Ross, Ronne-Filchner and Amery ice shelves show very low mean values for all $\gamma_0$ values. We do not need to cool down our basin to obtain a stable ice sheet. If anything, the thermal forcing correction is positive and can be higher than 1ºC in areas in which ice shelves extend beyond observations. This is due to the fact that we do not artificially limit the extension of our ice shelves. In order to deepen our understanding of this issue, we have done additional simulations limiting the ice-shelf extension, as is done in other studies. When limiting the ice shelves to the present-day observed extension the thermal forcing correction on the basins decreases. We will include this discussion in the new version of the article.

[Figure]

Figure A1*. Same as Figure A1 but with limited-ice-shelves.

8. As you are separating WAIS and EAIS in your analysis, and you are not showing 2-d representation of the forcings, I would suggest you add a figure which separates the averaged time series of your forcing for these 2 regions (or mention that the mean forcing evolutions are very similar in both regions).

We have modified Figure 1 to  include this  (see Figures section below). Indeed, the difference between the two regions is small, generally well below 1 ºC.

9. It is highly unlikely that the thermal forcing provided by the dataset will always coincide with an ice draft point in a cavity. Please describe your interpolation/extrapolation process in this case.

In order to interpolate the values at different depths, for oceanic fields like salinity and temperature (and therefore thermal forcing), Yelmo has a *marine-shelf interface*. Firstly, it computes the depth of the ice shelf base by distinguishing

between floating $\left( \text{depth}_{\text{shelf}} = H \cdot \dfrac{\rho_i}{\rho_{sw}} \approx 0.89 \cdot H \right)$ and grounded ice $(\text{depth}_{\text{shelf}} = z_{sl} - z_{bed})$ where $H$ represents the ice thickness, $\rho_i$ and $\rho_{sw}$ the densities of ice and sea water respectively, and $z_{sl}$ and $z_{bed}$ the height of sea level and bedrock. Secondly, the weights for the different vertical layers are calculated by taking into account the two nearest layers of the depth of the shelf previously calculated in the first step. Finally, the oceanic fields are calculated in terms of these weights as a weighted sum.

In the updated manuscript we will ensure that this description appears in an Appendix section.

10. In Figure 4, please add the grounding line location on your plot. Also, since you are focusing on WAIS in your results with ocean TF, I would appreciate a closer look at the WAIS which better shows your grounding line locations and your fit to observation. Currently, it is difficult to distinguish.

We have added the grounding-line location and the figure suggested with a close-up into the WAIS (see the Figures section below). We will incorporate the modified figure plot in the new version of the manuscript.

11. Your inversion procedure results in a lot more ice shelves than currently observed. Please discuss the impact of this feature on your results. For example, in the Amundsen Sea Embayment (ASE) your model predicts an ice shelf that could provide more buttressing compared to current observations.

Simulating the correct ice extent of ice shelves is a challenging task. Some ice models impose fixed ice-shelf extensions or prevent ice-shelves to grow beyond the observed ice front (Seroussi et al., 2019). In addition, tuning parameters to simulate realistic ice-shelf extensions may differ between different Antarctic embayments (Wilner et al., 2023). In effect, we allow ice shelves to extend further than observed ice-shelf fronts, which is especially noticeable for the Ronne-Filchner and the Ross ice shelves.

For the ASE, indeed the ice shelf covers all the area, advancing various kilometres into the ocean and that results in a deceleration of some ice streams flowing towards that sector at the beginning of the simulations (see Fig. 4 for a comparison with observations). As mentioned before, higher values of $\gamma_0$ results in a more realistic set up at the beginning (which can be observed in terms of RMSE, Fig. 3), despite having a greater rate of melting.

We have done additional simulations considering limited ice shelves with the mask from observations in the BedMachine dataset. In terms of sea-level contribution, results and spreads are very similar for higher-emission scenarios (Fig. A3) with differences ranging between 0.4 and 23.5% in absolute value, but generally being less than 10%. In the case of the low-emission simulation this

difference is much greater. In general we can say that our extended simulations have a considerable effect on sea level, but less than 141 mm SLE at 2500.

|  | CCSM4 | HadGEM2-ES | CESM2-WACCM | UKESM1-0-LL | UKESM1-0-LL (low) |
|---|---|---|---|---|---|
| $\gamma_0$=9620 m/yr | 23.5 | -10 | 0.4 | 1.2 | 38 |
| $\gamma_0$=14500 m/yr | 13.5 | 4.3 | 1.9 | 4.7 | -48 |
| $\gamma_0$=21000 m/yr | 7.9 | 3.1 | 0.9 | 2.7 | -4.9 |

Table A2. Differences (in percentage) in sea level contribution at 2500 for the experiments carried out, between the runs with extended ice shelves and the ones with limited ice shelves.

**Sea Level Contribution (mm SLE)**

[Figure]

a)

| | |
|---|---|
| CCSM4: | 498 (-243,1090) mm |
| HadGEM2-ES: | 3259 (2102,3854) mm |
| CESM2-WACCM: | 3505 (2701,4033) mm |
| UKESM1-0-LL: | 2769 (1598,3561) mm |
| UKESM1-0-LL (low): | 61 (-163,278) mm |

b)

| | |
|---|---|
| CCSM4: | 431 (-186,1005) mm |
| HadGEM2-ES: | 3119 (2313,3736) mm |
| CESM2-WACCM: | 3439 (2690,3996) mm |
| UKESM1-0-LL: | 2639 (1579,3464) mm |
| UKESM1-0-LL (low): | 90 (-101,292) mm |

c)

| | |
|---|---|
| CCSM4: | 67 (-57,86) mm |
| HadGEM2-ES: | 141 (-211,118) mm |
| CESM2-WACCM: | 66 (11,38) mm |
| UKESM1-0-LL: | 130 (19,98) mm |
| UKESM1-0-LL (low): | -29 (-62,-14) mm |

Figure A3. Evolution of the sea level contribution until 2500 for the different experiments carried out, together with the spread produced by the different values of $\gamma_0$. In a) with the extended ice shelves, in b) with the ice shelves limited by a mask from observations in the BedMachine dataset and in c) difference between a) and b).

We plan to include a more profound discussion of this matter in the Discussion section, and show this figure on the Appendix.

12. In your text, you refer to specific location with TF warming (e.g., ASE and Bellingshausen line 226) but point to an averaged time series as "proof". For this reason, I would suggest adding a figure showing this TF in 2015 and 2300. For

practicality, you could show the depth averaged TF between depth raging 200 m to 800 m, say (where the groundling line is located mainly).

In the revised manuscript , we will add the following new figure in the Appendix (Fig. A4).

[Figure]

Figure A4. Depth-averaged thermal forcing under the ice shelves (ºC) between 210 and 810 metres in (a) 2015 and in (b) 2300.

13. Please show in one of your figure or mention in your text the behavior of your control run over the length of your experiment. You mention that you are running one, but do not discuss its evolution (even one sentence could suffice).

The control runs with the different values of $\gamma_0$ are all stable during the time of simulation, with variations rounding 5 mm of SLE between 2015 and 2500 (Fig. A5) in terms of volume above floatation (m SLE).

With increasing values of $\gamma_0$, the ice sheet has less ice volume at its initial state, but nevertheless variations are negligible ranging between 10-15 cm SLE (Fig A5).

[Figure]

Figure A5. Evolution of the volume of sea level equivalent (m SLE) in the control runs for the AIS with the different values of $\gamma_0$.

In Fig. 1, you go through the trouble of differentiating the TF at the grounding line from the one in cavities. While for most model the TF at the grounding line is always warmer compared to in the cavity by a fraction of a degree, the TF from CESM2-WACCM is warmer by a couple of degrees in the cavities. Could you infer any importance on this behavioural difference from your results?

There may be some confusion here. We are not differentiating between TF at the grounding line from TF in the cavities. We are making a comparison at the grounding line for panels b and c in Figure 1. The difference comes when calculating the curves. For Fig. 1b, we take the data from ISMIP6 masking on the grounding line of 2015 (which is fixed). For Figure 1c, we consider Yelmo outputs of TF on the ice shelves and let the mask evolve while evolving the grounding line (retreating more precisely). So that is what leads to a difference, while for Fig. 1b, the grounding line is fixed for all models, in Fig. 1c, it evolves differently in each case. For CESM2-WACCM, it is especially noticeable because it retreats more towards the interior.

In my (limited) experience about the use of YELMO in publication, it is more frequently used in paleoclimate studies as opposed to century forecasting ones. In this study YELMO is run at a resolution of 16km which is a resolution many people would argue is too coarse to help reduce uncertainties in future projections. If you can, I would suggest you try to make the point in arguing that your results are encouraging at that resolution. I think you started hinting at this with your table 4. I would also suggest adding the results from Lipscomb et al. (2023) to this table and add the resolution of each model for their study. Also, as hinted above, you could succinctly (or not) write a section about your results regarding the ISMIP6 experiment until 2100 and compare YELMO to the output from Seroussi et al. (2020). This would be of limited extra work since these would be a subset of your current runs.

Indeed, Yelmo has often been used for paleo-simulations with timescales of several thousand years (Blasco et al. 2021, Moreno-Parada et al. 2022). Concerning horizontal resolution, for this work we first ran all simulations with a 32 km resolution. This was very informative as to what to expect from the model, and then we moved forward with 16 km resolution simulations, which helped in better simulating the present-day ice sheet with Yelmo.

At 32 km, results for sea level evolution are very similar, varying 7-10% for high-emission scenarios (Fig. A6). Generally, they yield an increase in the contribution but a decrease in the spread between the low and high values of $\gamma_0$ (Table A3). Lipscomb et al. (2021) made the point with 2 and 4 km resolution with sea level variations of ~15% and with 4 and 8 km with ~20%, but they find that the higher the resolution, the higher the sea-level contribution. In our case, it is the contrary, at least at the resolutions we tested. We will add results from Lipscomb et al. (2021) to Table 4 in our manuscript (see section Tables below).

| | CCSM4 | HadGEM2-ES | CESM2-WACCM | UKESM1-0-LL | UKESM1-0-LL (low) |
|---|---|---|---|---|---|
| **16 km** | 1333 | 1752 | 1332 | 1963 | 441 |
| **32 km** | 1199 | 1458 | 1357 | 1810 | 356 |

Table A3. Comparison with different resolutions of the spread (mm SLE) at 2500, between the low and high value of $\gamma_0$ for the experiments carried out.

[Figure]

Figure A6. Sea level contribution for the different experiments carried out and spread of the values of $\gamma_0$ using resolutions of 16 km (a) and 32 km (b). In c, differences between a and b.

Sutter et al. (2023) argued using PISM with 16, 8 and 4 km resolution grids that their 8 km ice-sheet configuration is close to their 16 km runs, nonetheless the rate of retreating at the grounding line is different, starting sooner for 8 and 4 km compared with 16 km, especially observed for Thwaites Glacier. This suggests the MISI instability is triggered earlier when using a higher resolution. Yet, running simulations at higher resolution (8 km or finer) is currently technically infeasible for Yelmo, as we do not have a parallelized procedure and it would be computationally very expensive.

Concerning the comparison with Seroussi et al (2020) until 2100, as shown in Figure 12 from that paper, the spread and mean values produced by the different ice-sheet models ranges between -23 and 35 mm SLE for the medium value of $\gamma_0$ (Figure A7). Our results for 2100, are similar but only three GCM models fit within the former spread: CESM2-WACCM and the high and low emission cases with UKESM1-0-LL, with a negative contribution in 2100. HadGEM2-ES exceeds in approximately 25 mm the upper limit from Seroussi et al. (2020) and CCSM4 does not reach the lower limit being 12 mm below it. Regarding the low value, only HadGEM2-ES fits in the range of the shading while all the other forcings are below the lower limit during the whole period (Figure A7a). Finally, with respect to the high value, only HadGEM2-ES diverges from the shading from 2070 and on, while the others are enclosed in most years of the period. A possible explanation for this can be based on the fact that the forcings evolution (Fig. 1) is very small during the first 100 years, expanding from that year and on. Also, we are using two CMIP6 models while in Seroussi et al. (2020), all the forcings come from CMIP5 models.

For the reviewed manuscript, we will include a paragraph to justify our choice of horizontal resolution and include Fig. A7 as Supplementary material to have a comparison with Seroussi et al. (2020) results, showing the ranges where sea level contributions by Yelmo are.

[Figure]

Figure A7. Sea level contribution (mm SLE) for the experiments carried out with five GCM forcings under high-emission scenarios, until the year 2100. In a) the low value of $\gamma_0$=9620 m/yr, in b) the medium value $\gamma_0$=14500 m/yr and in c) the high value $\gamma_0$=21000 m/yr. Solid lines represent the curve for the values of $\gamma_0$ and the shading the overall results from Seroussi et al. (2020) with different ice-sheet models and the corresponding value for $\gamma_0$.

I am reiterating the significance of \deltaT_sector in your results interpretation. Particularly, when interpreting the results for CCSM4 (e.g., Sect. 4, lines 924-296). (Since I have not seen the values for \deltaT_sector, this comment is somewhat speculative but with a point to be addressed nonetheless). If your inversion leads to a \deltaT_sector=-3 degC in the ASE and the TF for CCSM4 is about 4 degC, your model will actually feel only 1degC there. This means that the slow response of say Thwaites,

is due to your model calibration as opposed to the forcing itself. This is a particular and important point to make to not undermine the importance of a 4 degC TF suggested by CCSM4. Please clarify this point in your discussion. As a parallel, I would refer to Lipscomb et al. (2021) and their 2 degC synthetic experiment in the ASE for that exact reason.

As mentioned above and shown in Fig. A1, this is not the case: thermal forcing corrections are generally below 1.5ºC in most basins of the WAIS and EAIS. With the limited-ice-shelves simulations and repeating the analogue for Figure A1, we can infer a reduction of this warming correction.

The beginning of section 3.1 refers to the initialization procedure's results. I would move it at the end of section 2.3 or create a new section 2.5. In any event it is out of place in a section that is supposed to introduce sensitivity experiments.

In order to make it more clear, we will put the initialization results in a section 2.5 addressing the discussion on this.

Throughout your manuscript you refer to sea level equivalent in your figures. I did not see anything indicating otherwise, therefore I presume that the values were converted solely from the ice mass above floatation. I would like to bring to your attention that this conversion is not as simple (Goelzer et al. (2020)) and I would encourage you to document your conversion to avoid future readers' ambiguity. Along these lines, please add "s.l.e" (sea level equivalent) to "m" when you refer to m of sea level in your units in the text.

Thank you for this important comment. Regarding the conversion, yes, Yelmo returns the variable "V_sle" which calculates the volume of ice above flotation for grounded ice in metres. The conversion is done based on the ice above floatation because as Goelzer et al. (2020) pointed out, on short timescales, this conversion can assume a constant area for the ocean when calculating the SLC. Therefore, in a five-century period the bedrock is not expected to change considerably. In this study Yelmo uses ELRA as GIA model. Seroussi et al. (under review) have demonstrated (pers. comm.) that an ELRA model correction in the bedrock elevation could result in differences of around +11-17% per ~1.3 m SLE calculated with the volume above floatation method. Our results are conservative, but nonetheless we are mainly focused on sensitivities in our work rather than absolute contributions. To make the final computation shown in the evolution sea level curves in the text, we take into account the control run as reference. We will also add SLE to m or mm in the revised text as we have put in our figures.

In the near future, Yelmo will incorporate a more realistic GIA model (FastIsostasy, https://github.com/JanJereczek/FastIsostasy.jl, Swierczek-Jereczek et al., in revision) that could be interesting to test with these ISMIP6-2300 simulations. Anyhow, we will make sure to document the conversion in the new version.

Regarding the figures, I found the spatial figures really hard to read and would highly suggest enhancing them by making them bigger for a start.

We have remade them bigger and with grosser linewidths for coastlines and grounding lines.

Data availability: while the YELMO code is open source, please provide a link to your output and configuration files for reproducibility purposes.

We have created a Zenodo repository with a DOI identification (10.5281/zenodo.10657938 ) containing a zip file with different folders with the NetCDF Yelmo output files for 1D and 2D variables for each experiment and value of $\gamma_0$. We will publish it upon acceptance.

**Technical comments**

For the technical comments below, we will take special care to address each one for the next version.

Line 52: please cite Edwards et al. (2021) regarding ice-ocean interaction uncertainty. We will do so.

Line 68: remove the http link and web address from the text. We will remove it.

Line 75: please include results from Lipscomb et al. (2021) in your discussion here. We will do so.

Line 89: please cite Nowicki et al. (2020), Jourdain et al. (2020), and Seroussi et al. (2024) for the extended ISMIP6 AIS 2300 dataset. (See main comments above.). Remove the http citation. We have contacted the ISMIP6-2300 team leaders to inform them about our publication and to ask for the proper citation.

Line 109: please cite Goldberg et al. (2011) as a reference to DIVA. We will cite it.

Line 110: The description of the AIS grid is in a weird location in the text as it precedes the reference to idealized experiments. I suggest moving this detail at a less ambiguous place (maybe end of paragraph (see next comment)). We will follow your suggestion and move the description of the grid to the end of the paragraph, also reorganising with your following comment.

Line 114: I would start a new paragraph after "Last Glacial Maximum". We will reorganise these lines.

Line 119: I believe you mean Leguy et al. (2014) (their 2013 manuscript was in the TCD and the 2014 version is the final published version). Yes, we will revise that.

Line 123: here and elsewhere I would suggest using a more active voice instead of passive. I would replace "has been" by "is". Thank you, we will check not only this line but also the rest of the text.

Line 143: remove the reference to the wiki page. We will do it and add a correct citation.

Line 152: "bellow the rest" please be more precise. Thank you for pointing this out, we will rephrase it in the next version of the manuscript.

Line 152-153: "Therefore […]" I find this sentence odd, out of place, erroneous, and not adding any useful information to the manuscript. I suggest you remove it, or you clarify what you are trying to get at. You are trying to compare the output of global climate models (GCMs) that are (commonly) tuned for the pre-industrial time period and then react to a forcing scenario after the historical period. The atmospheric and oceanic forcing used here are a response of the GCM to SSPs and RCPs forcings. When I read your sentence, it almost sound to me that you are trying to say that the atmospheric and oceanic forcings co-evolve independently which is simply not true.

We will change the sentence for another which makes this point more clear.

Line 159: replace "On the shelves" by "in ice shelf cavities" (or "In cavities)". Thank you for the correction.

Line 160: what are the "main" cavities of WAIS and EAIS? Please be more precise. We will be more clear in the new manuscript.

 Line 161: see comment for line 159.

Line 161: "The grounding line […]" I don't understand the point you are trying to make here. Please remove or rephrase. We will rephrase it in the reviewed text version.

Line 164: here and elsewhere, I find it clearer to refer to equations as "(Eq. 3)" compared to "(3)". Thank you, we will change it along the reviewed version.

Line 169: where is the climatology from? Note that Bedmachine is a dataset for bed topography, ice thickness and surface elevation, not for ice velocity, SMB, or air temperature.

For the atmospheric fields, Surface Mass Balance (SMB) and 2m-air temperature, we have used monthly values from the Regional Atmospheric Climate Model (RACMO v2.3) together forced by ERA-Interim reanalysis mean climatology dataset from 1981-2010. In the submitted version there is a mistake on our side when saying that we use a reference climatology average in the period 1995-2014. Regarding the oceanic fields, the reference is Jourdain et al. (2020). We will clarify this in the final version.

Line 170: what happens during the last 5kyrs of the initialization's procedure?

During the first 15 kyr, the basal friction and thermal forcing fields are optimized.  During the last 5 kyr, the optimised  fields are held constant and the spin-up is run with the reference climatology towards steady state.  We will make this point clear in the revised text. This will be clarified in the revised text.

Line 179: please provide a map for \DeltaT_sector for the 3 \gamma0 parameters.

It is now provided as Figure A1; see the discussion above.

Line 188: I would suggest rewriting this sentence with an active voice: "conducted: atmosphere-only runs in which […] thermal forcing are imposed […] simulation and ocean-only simulations with the surface mass balance and surface temperature […]"

Thank you for the suggestion, we will change to active voice along the text.

Line 189: (regardless of my previous comment) remove "temperature and salinity" since the thermal forcing was derived using these 2 variables. Yes, this will be deleted for the new version.

Line 190: why do you feel the need in mentioning both precipitation and SMB? We will change it in the text.

Beginning of Section 3.1: see main comments about the discussion of initialization.

In order to make it more clear, we will put the initialization results in a section 2.5 addressing the discussion on this.

Line 193: what dataset are you using for ice velocity comparison?

The dataset is Rignot et al. (2011), as mentioned above. This will be properly stated in the revised text.

Line 194: there is no need to keep referring to Bedmachine and its reference. Once is enough. We will correct for this.

Line 198: similarly to above, I would suggest a more active voice and rewrite as: "For the medium value of \gamma0, the model overestimates ice thickness at the ice-sheet margins." Thank you, we will proceed as suggested.

Line 202: please rephrase. We will do it in the next version.

Line 203: replace "the values of" by "varying". We will do it in the next version.

Line 205: 3.5 m of sea level (or s.l.e after defining the acronym somewhere). Yes, we will use an acronym previously defined.

Line 206: replace "but" by "and". We will do it in the next version.

Line 209: replace "over" by "under". We will do it in the next version.

Line 210: I believe you meant to refer to Fig 1.c. We will revise this in the text and clarify it.

Line 211: say more about the difference in temperature and the rate of change in the first century. We will do it and expose it clearly in the new manuscript.

Line 212: remove "Furthermore". We will do it in the next version.

Line 220: the contribution of WAIS and EAIS adds up to 95%. Where are the 5% missing, from the control?

No, it was just approximated numbers from two GCM models (CESM2-WACCM and UKESM1-0-LL) on Table 3. In other words, we just try to summarise Table 3 with some numbers.

Line 220: replace "1 m" by "\approx 1 m s.l.e". Yes, we will use an acronym previously defined.

Line 221: replace "thus […] even" by "being the major contributor". We will do it in the next version.

Line 227: Fig 1 is not a spatial figure and for this reason it does not support your claim for the warmer waters of the ASE and Bellingshausen. Yes, we will consider another figure, like you have proposed before and rephrase this.

Line 236-237: you have not defined which are the main ice shelves of the west (this is ambiguous). Remove "of the main ice shelves of the west". Certainly, we have not. Therefore, we will remove that sentence.

Line 284: replace "not allow" by "not lead". Also, you have not shown spatial patterns of SMB. Maybe it might be worth doing so. We will replace the words you say and consider the showing of SMB.

Line 291: the relationship of \gamma0 on sea level contribution was also shown in Berdahl et al. (2023). Please add this to your discussion. We will ensure to do it in the Discussion section as previously mentioned at the beginning of the reply.

Line 336: update citation and remove the web address. Yes, we will do it.

**Tables**

Table3: in caption, replace "at the year" by "at year" and "the heat exchange velocity" by "\gamma_0". We will do it in the new revised text.

| GCM Model | Chambers et al. (2021) / 8 km | | Greve et al. (2022) / 8 km | Lipscomb et al. (2021) / 4 km | Yelmo / 16 km | |
|---|---|---|---|---|---|---|
| | 2300 | 2500 | 2300 | 2500 | 2300 | 2500 |
| CCSM4 | 0.5 | 2.1 | 1.25 | 0.28 | 0.1 | 0.5 |
| HadGEM2-ES | 1.6 | 3.6 | 3.3 | 0.35 | 1.5 | 3.2 |
| CESM2 | 0.45 | 1.75 | 0.95 | 0.26 | 1.7 | 3.5 |
| UKESM1-0-LL | 1 | 2.6 | 1.95 | 0.38 | 1.2 | 2.8 |

Table 4. Sea-level contribution (m SLE) at 2300 and 2500 among the studies made by Chambers et al. (2021), Greve et al. (2022), Lipscomb et al. (2021) with different resolutions and the results from Yelmo.

**Figures**

We here provide the new versions of the figures following the referee's suggestions. We will replace them in the text and add some others that appear before on this reply.

Figure 1: In your text, you mention that the EAIS and WAIS will be affected differently by the forcing. It might be worth it to add the mean forcing for EAIS and WAIS separately in this figure.

Also:

1. Panel b: add "at the grounding line" to the y-axis label.
2. Panel c: how did you obtain your thermal forcing, did you filter them out to only account for year 2015 cavities or are these averages over each full basins in your Antarctic grid? Please add some details here.

Yelmo outputs the evolution in time of the thermal forcing over the shelves. To produce Fig. 1c, we simply mask on the grounding lines points which are evolving, opposite to Fig. 1b where the grounding line is fixed in 2015. We will provide a clarification on this issue in the new text.

3. For each panel and the figure caption, add that these are mean evolutions.

[Figure]

Figure 1. Mean evolution of the atmospheric (a), oceanic thermal forcing at the grounding line depth fixed in 2015 (b) and thermal forcing at the evolving grounding lines in the AIS (c) for the five experiments carried out: CCSM4 (RCP8.5), HadGEM2-ES (RCP8.5), CESM2-WACCM (SSP5-8.5), UKESM1-0-LL (SSP5-8.5) and UKESM1-0-LL (SSP1-2.6). For (d), (e) and (f), same as in (a), (b) and (c) respectively but for the WAIS and EAIS. From 2300 to 2500 a constant climate is imposed through the average climatic conditions in the period 2291-2300.

Figure 2: Please make the figure bigger. Also:

1. Add labels to the colorbars.
2. Add a legend to the figure describing the grounding lines.
3. In figure caption, I believe (b) shows thermal forcing differences.

[Figure]

a)

| CCSM4 | HadGEM2-ES | CESM2-WACCM | UKESM1-0-LL | UKESM1-0-LL (low) |

b)

- - - - 2015 Grounding Line  ——— 2015 Coastlines  ——— 2300 Grounding Line

Figure 2. Surface temperature anomaly (a, in ◦ C) between 2300 and the present day as simulated by each of the GCMs selected from ISMIP6-2300 in Table 1, using four high-emission scenarios and a low emission one. Thermal forcing differences over the ice shelves (b, in ◦ C) between 2300 and 2015, together with the grounding lines and coastlines at the initial time and the 2300 grounding line.

Figure 3: I would suggest creating 2 numbers, one with your extended ice shelves, and one that uses the same ice extend as Bedmachine.

[Figure]

Figure 3. Root mean square error (RMSE) for ice thickness (in m) and ice surface velocity (in m/yr) between the start of the simulations in 2015 and observations. The different colorbars represent the values of $\gamma_0$. To mask over the cells containing ice, the fields at the start of the simulation (extended ice shelves) have been used for the columns labelled with (1) and the observations for the ones labelled with (2).

Figure 4:

1. Add titles to the figures.
2. Create a subplot zooming in on the WAIS.
3. Add the grounding line locations for both model and obs.
4. What data did you use for ice velocity? (Note that Bedmachine used a dataset and if you are using the same, you still need to refer to it separately.)

[Figure]

Figure 4. Ice thickness (a, in m) and surface velocity anomalies (b, in m/yr) at 2015 for the medium value of $\gamma_0$, with respect to observations from the BedMachine Antarctica V2 dataset and Rignot et al. (2011) dataset respectively. Zoom over the WAIS region (c and d) to better see details on the grounding line.

Figure 5:

1. In the caption, line 2, replace "The values on" by "The values in".
2. In the caption, line 3, add "at year 2500" after "contribution".

[Figure]

Figure 5. a) Projections of the AIS sea-level contribution (mm SLE) for the five scenarios chosen (Table 1), relative to the control run. In shaded colours the spread resulting from the range of $\gamma_0$ values is represented; the solid line corresponds to the medium value. The values in the legend are indicative of the sea-level contribution given by the medium value followed inside the parentheses by the corresponding low and high value. b) Same as a) but for the rate of sea level contribution (mm SLE/yr).

Figure 6:

1. Place the title from the y-axis to the top of the figure.
2. Add a y-axis title "WAIS" for panel (a) and "EAIS" for panel (b).

[Figure]

Figure 6. Evolution of the sea-level contribution (mm SLE) from the two main regions of Antarctica, the WAIS (including the Antarctic Peninsula, a) and the EAIS (b), relative to the control run, since the start of the simulations in 2015 to year 2500.

Figure 7:

1. Make this figure bigger if you can.
2. It is very difficult to see the grounding lines. It looks like the stronger differences are for WAIS so maybe create a figure that focuses on WAIS.

[Figure]

Figure 7. Ice thickness anomalies (in m) for the years 2100, 2200, 2300, 2400 and 2500 with respect to the start of the simulations for the set of five experiments carried out with the median value of $\gamma_0$. The grounding line is represented by black and violet colours for 2500 and 2015 respectively. To show negative anomalies in the WAIS, no masking has been implemented as time evolves, keeping the original coastlines.

[Figure]

Figure 7*. Ice thickness anomalies (in m) in the WAIS (defined as in Fig. 4a) for the years 2100, 2200, 2300, 2400 and 2500 with respect to the start of the simulations for the set of five experiments carried out with the median value of $\gamma_0$. The grounding line is represented by black and violet colours for 2500 and 2015 respectively. To show negative anomalies in the WAIS, no masking has been implemented as time evolves, keeping the original coastlines.

[Figure]

Figure 8. As Figure 7 but for ice surface velocity anomalies (in m/yr). Note that only the cells where there is ice (floating and grounded) at the specific times are represented.

[Figure]

Figure 8*. As Figure 7* but for ice surface velocity anomalies (in m/yr).

Figure 10:

1. Do these time series begin with the Yelmo ice extent which is larger than observed?

Yes, the initial state of the simulations is the last state of the spin-up which in fact has a larger than observed ice extent in some ice shelves around Antarctica, especially in the west region but also in the east. For the revised manuscript, we will write a sentence at the caption or in the text explaining this.

Figure 11:

1. Make this figure bigger if you can.

[Figure]

Figure 11. Ice surface velocity evolution (in m/yr) at the beginning of each century for the experiments with the GCM models CCSM4 (top) and CESM2-WACCM (bottom). Solid lines represent the coastline and surface elevation levels at the year considered while dashed lines delineate grounding lines.

Figure 12:

1. Place the title from the y-axis to the top of the figure.
2. Add a y-axis title "Atmospheric forcing only" for panel (a) and "Ocean forcing only" for panel (b).
3. In figure caption, instead of saying "is neglected […]" maybe say something like "the inversion process lead to similar initial ice sheet states regardless of \gamma_0". (\gamma_0 influences the impact of ocaen state in ice cavities.)

[Figure]

Figure 12. Sea-level contribution of the AIS (mm SLE) in the cases where only the forcing of the atmosphere (a) or the ocean (b) are considered. The shading in b) indicates the spread produced by the effect of the values of $\gamma_0$, while in a), as the optimization process leads to similar initial ice sheet states regardless of $\gamma_0$, the ocean fields maintain the present-day state.

Figure 13:

1. Make the figure and the font size bigger.
2. Refer to the panels in your caption.

[Figure]

Figure 13. Ice-thickness anomalies (in m) for the atmosphere-only (a) and ocean-only (b) cases between the year 2500 and the start of the simulations. Solid lines and dashed lines represent the coastline and grounding line, respectively, at year 2500. Numbers included in the maps represent the total sea level contribution in metres for the medium value of the basal melting parameter.

**References** We will add them to the next version and cite them properly. Note that there are some others in blue cited by us in our answer.

Barthel, A., Agosta, C., Little, C. M., Hattermann, T., Jourdain, N. C., Goelzer, H., Nowicki, S., Seroussi, H., Straneo, F., and Bracegirdle, T. J.: CMIP5 model selection for ISMIP6 ice sheet model forcing: Greenland and Antarctica, The Cryosphere, 14, 855–879, https://doi.org/10.5194/tc-14-855-2020, 2020.

Berdahl, M., Leguy, G., Lipscomb, W. H., Urban, N. M., & Hoffman, M. J. (2023). Exploring ice sheet model sensitivity to ocean thermal forcing and basal sliding using the Community Ice Sheet Model (CISM). *The Cryosphere*, *17*(4), 1513-1543.

Blasco, J., Alvarez-Solas, J., Robinson, A., and Montoya, M.: Exploring the impact of atmospheric forcing and basal drag on the Antarctic Ice Sheet under Last Glacial Maximum conditions, The Cryosphere, 15, 215–231, https://doi.org/10.5194/tc-15-215-2021, 2021.

Edwards, T., Nowicki, S., Goelzer, H., Seroussi, H., Marzeion, B., Smith, C. E., . . . Gladstone, R. (2021). Quantifying uncertainties in the land ice contribution to sea level rise this century. Nature, 593 . doi: 10.1038/s41586-021-03302-y

Goelzer, H., Coulon, V., Pattyn, F., De Boer, B., & Van De Wal, R. (2020). Brief communication: On calculating the sea-level contribution in marine ice-sheet models. *The Cryosphere*, *14*(3), 833-840.

Goldberg, D. N.: A variationally derived, depth-integrated approximation to a higher-order glaciological flow model, J. Glaciol., 57, 157–170, https://doi.org/10.3189/002214311795306763, 2011.

Jourdain, N. C., Asay-Davis, X., Hattermann, T., Straneo, F., Seroussi, H., Little, C. M., and Nowicki, S.: A protocol for calculating basal melt rates in the ISMIP6 Antarctic ice sheet projections, The Cryosphere, 14, 3111–3134, 2020.

Leguy, G. R., Asay-Davis, X. S., & Lipscomb, W. H. (2014). Parameterization of basal friction near grounding lines in a one-dimensional ice sheet model. The *Cryosphere*, 8(4), 1239-1259.

Lipscomb, W. H., Price, S. F., Hoffman, M. J., Leguy, G. R., Bennett, A. R., Bradley, S. L., ... & Worley, P. H. (2019). Description and evaluation of the community ice sheet model (CISM) v2. 1. *Geoscientific Model Development*, *12*(1), 387-424.

Lipscomb, W. H., Leguy, G. R., Jourdain, N. C., Asay-Davis, X., Seroussi, H., and Nowicki, S.: ISMIP6-based projections of ocean-forced Antarctic Ice Sheet evolution using the Community Ice Sheet Model, The Cryosphere, 15, 633–661, 2021.

Moreno-Parada, D., Alvarez-Solas, J., Blasco, J., Montoya, M., and Robinson, A.: Simulating the Laurentide Ice Sheet of the Last Glacial Maximum, The Cryosphere, 17, 2139–2156, https://doi.org/10.5194/tc-17-2139-2023, 2023.

Morlighem, M., Bondzio, J., Seroussi, H., Rignot, E., Larour, E., Humbert, A., & Rebuffi, S. (2016). Modeling of Store Gletscher's calving dynamics, West Greenland, in response to ocean thermal forcing. *Geophysical Research Letters*, *43*(6), 2659-2666.

Nowicki, S. M., Payne, A., Larour, E., Seroussi, H., Goelzer, H., Lipscomb, W., Gregory, J., Abe-Ouchi, A., and Shepherd, A.: Ice sheet model intercomparison project (ISMIP6) contribution to CMIP6, Geoscientific Model Development, 9, 4521–4545, 2016.

E. Rignot *et al.*, Ice-Shelf Melting Around Antarctica.*Science* 341, 266-270 (2013). DOI:10.1126/science.1235798

Seroussi, H., Nowicki, S., Simon, E., Abe-Ouchi, A., Albrecht, T., Brondex, J., ... & Zhang, T. (2019). initMIP-Antarctica: an ice sheet model initialization experiment of ISMIP6. *The Cryosphere*, *13*(5), 1441-1471.

Seroussi, H., Nowicki, S., Payne, A. J., Goelzer, H., Lipscomb, W. H., Abe-Ouchi, A., Agosta, C., Albrecht, T., Asay-Davis, X., Barthel, A., et al.: ISMIP6 Antarctica: a multi-model ensemble of the Antarctic ice sheet evolution over the 21st century, The Cryosphere, 14, 3033–3070, 2020.

Seroussi, H., et al.: Evolution of the Antarctic Ice Sheet over the next three centuries from an ISMIP6 model ensemble, Earth's Future, 2024 (under review)

Shapiro, N. M. and Ritzwoller, M. H.: Inferring surface heat flux distributions guided by a global seismic model: particular application to Antarctica, Earth Planet. Sci. Lett., 223(1-2), 213–224, doi:10.1016/j.epsl.2004.04.011, 2004.

Sutter, J., Jones, A., Frölicher, T. L., Wirths, C., & Stocker, T. F. (2023). Climate intervention on a high-emissions pathway could delay but not prevent West Antarctic Ice Sheet demise. Nature Climate Change, 13(9), 951-960.

Swierczek-Jereczek, J., Montoya, M., Latychev, K., Robinson, A., Alvarez-Solas, J., & Mitrovica, J. (2023). FastIsostasy v1. 0–An accelerated regional GIA model accounting for the lateral variability of the solid Earth. *EGUsphere*, *2023*, 1-34.

Wilner, Joel & Morlighem, Mathieu & Cheng, Gong. (2023). Evaluation of four calving laws for Antarctic ice shelves. The Cryosphere. 17. 4889-4901. 10.5194/tc-17-4889-2023.

---

## Author Comment (AC2)

**Review 2 of Antarctic sensitivity to oceanic melting parameterizations**

Juarez-Martinez et al. investigated the impacts of emission scenarios, the climate forcings and the heat exchange velocity in the basal melt parameterisation on the projected Antarctic mass loss before 2500 They found that the Antarctic ice sheet contributions to sea level rise highly depend on the heat exchange velocity at the ice-ocean interface and also the climate forcing chosen at high emission scenario. Separate atmospheric-only and oceanic-only experiments show that oceanic forcing plays a dominant role for the West Antarctic while atmospheric forcing is more important for the eastern sector and the interior. Overall, the manuscript is well written. However, the model setup section and some of the descriptions on results need improvements. I'm happy to support the publication of this study after major revisions.

We are thankful to the referee for their helpful feedback. We will take special care in improving the model setup section and the description of the results, and in implementing all the suggestions in the new version of the manuscript. Note that our comments are in blue.

**Here are some general comments:**

In the model description, some information is missing, including how the ice front position is updated, how the basal drag is treated for partially-floating cells, the reference SMB dataset. About the moving ice front positions, how the ice front position is updated is not described at all. Is a calving included? Could you please show the difference between the ice front position after spin-up and the present- day? How would the difference affect your results?

Yes, calving is computed following a von Mises calving law following the approach in Lipscomb et al. (2019). This gives a calving-retreat rate which is included in the ice shelf front. If the ice front velocity is larger than the calving rate, the ice front advances; if it is lower, the ice front retreats. The parameters related to this law are similar to those used in Lipscomb et al. (2019). The scaling calving parameters are $k_t$=0.0025 m/(yr Pa), $w_2$=25 for the calving eigenvalue weighting coefficient, and there is a calving limit in ice thickness set to 200 m. Regarding the grounding-line melt parameterization, we have used the PMP parametrization, where melting is scaled with the floating fraction (Leguy et al., 2021). Therefore we are reducing the basal dragging in the grounding line in accordance with the effective pressure. Finally, for Surface Mass Balance (SMB) and air (2 m) temperature we have used monthly values from the Regional Atmospheric Climate Model (RACMO v2.3) forced by the ERA-Interim reanalysis mean climatology from 1981-2010. For the oceanic fields, we have used the dataset from Jourdain et al. (2020). They first combine two known ocean climatologies: a prerelease from September 2018 of the NOAA World Ocean Atlas 2018 dataset and the Met Office EN4 subsurface ocean profiles with the complementary MEOP data. After that, they extrapolate the oceanic data to calculate the data under the ice shelves for each sector of the 16 in which Antarctica is divided (see Section 3 in Jourdain et al.,

2020 for more details). We will include a more complete explanation of our setup as discussed here in the updated manuscript version.

We include below an updated figure (Figure 4) showing the AIS configuration after the spin-up (at the initial year, 2015) compared with observations. Following a request by another referee we zoom into the WAIS sector.

[Figure]

Figure 4. Ice thickness (a, in m) and surface velocity anomalies (b, in m/yr) in 2015 for the medium value of $\gamma_0$, with respect to observations from the BedMachine Antarctica V2 dataset and Rignot et al. (2011) respectively. c and d show insets of a and b, respectively, zoomed over the WAIS region.

After the spin-up, our model shows extended ice shelves compared with observations. The extension of the shelves is found to be dependent on the value of $\gamma_0$, increasing with decreasing $\gamma_0$ values. The highest $\gamma_0$ value yields the closest distribution to observations. This can also be seen from the RMSE values shown on Figure 3 on the manuscript.We note that we do not impose a cutoff at the ice-shelf front as other

models do since we use exactly the same model setup for paleo simulations of colder climates (e.g. Blasco et al. 2019; 2021).

To test the impact of the overestimated ice-shelf extension on the ice-sheet response we have carried out additional simulations with limited ice-shelf extension masked with observations from the BedMachine dataset. In terms of sea-level contribution, results and spreads are very similar for higher-emission scenarios (Fig. B1) with differences ranging between 0.4 and 23.5% in absolute value, but generally being less than 10% (Table B1). In the case of the low-emission simulation this difference is much greater. In general we can say that our extended simulations have a stronger effect on sea level, but less than 141 mm SLE at 2500. Furthermore, the spread differences between the lower and higher values of $\gamma_0$ are not too big (Fig. B1c).

| | CCSM4 | HadGEM2-ES | CESM2-WACCM | UKESM1-0-LL | UKESM1-0-LL (low) |
|---|---|---|---|---|---|
| $\gamma_0$=9620 m/yr | 23.5 | -10 | 0.4 | 1.2 | 38 |
| $\gamma_0$=14500 m/yr | 13.5 | 4.3 | 1.9 | 4.7 | -48 |
| $\gamma_0$=21000 m/yr | 7.9 | 3.1 | 0.9 | 2.7 | -4.9 |

Table B1. Differences (in percentage) in sea level contribution at 2500 for the experiments carried out, between the runs with extended ice shelves and the ones with limited ice shelves.

[Figure]

Figure B1. Evolution of the sea level contribution until 2500 for the different experiments carried out, together with the spread produced by the different values of $\gamma_0$. In a) with the extended ice shelves, in b) with the ice shelves limited by a mask from observations in the BedMachine dataset and in c) difference between a) and b).

We will include this discussion in the new version of the manuscript.

Coarse resolution like 16 km would largely affect the estimation of mass loss and also the movement of grounding line, which would influence the basal drag and basal

melting applied at the grounding line. However, nothing related with this has been discussed, which I think is important.

Although we did not include a discussion on this issue, we did test the sensitivity of our results to the horizontal resolution, albeit with a coarser 32 km resolution. For 32 km, the results in terms of the global sea level evolution are very similar, increasing by 7-10% in high-emission scenarios for the medium value of $\gamma_0$ (Fig. B2). Yet, the spread at 2500 (Table B2) is very similar. Sensitivity to horizontal resolution has also been found in other studies, at different resolutions. Using CISM and extending until 2500, Lipscomb et al. (2021) showed sensitivities of up to ~20% with respect to horizontal resolution, although they used higher horizontal resolution values between 2-8 km using the CISM model. Same model with a resolution of 4 km was used by Berdhal et al. (2023), who made a similar study to ours but not only varying $\gamma_0$ but also the parameter p from Leguy's parametrization, which in our case is constant (p=0.5) in all experiments. Sutter et al. (2023) also found a sensitivity of their results with respect to horizontal resolution using PISM with 4-16 km. Although they argued that their 8 km ice-sheet configuration is close to their 16 km runs, the grounding-line retreat started earlier with increasing resolution, especially in the Thwaites Glacier.

In our case, we do not have a parallelized procedure to proceed with 8 km and it would be computationally unaffordable. Therefore we haven't been able to test the simulations on these shorter timescales for 8 km. In the revised text, we will nevertheless include this discussion with the possible influence of horizontal resolution in our results and consider comparisons with the aforementioned studies. Variations of the specific values from Lipscomb et al. (2021) and Berdhal et al. (2023) are susceptible to have much difference to ours quantitatively based not only in the resolution used but also in the parameters and the way in which they extend towards 2500 (using the ISMIP6-2100 dataset), but nonetheless results in terms of sensitivity for the $\gamma_0$ parameter are still well founded.

| | CCSM4 | HadGEM2-ES | CESM2-WACCM | UKESM1-0-LL | UKESM1-0-LL (low) |
|---|---|---|---|---|---|
| **16 km** | 1333 | 1752 | 1332 | 1963 | 441 |
| **32 km** | 1199 | 1458 | 1357 | 1810 | 356 |

Table B2. Comparison of the spread (mm SLE) at 2500 between the low and high value of $\gamma_0$ for the experiments carried out for model resolutions of 16 and 32 km.

[Figure]

Figure B2.  Sea level contribution for the different experiments carried out and spread of the values of $\gamma_0$ using  resolutions of 16 km (a) and 32 km (b). In c, differences between a and b.

The  section  3.1  needs  to  be  restructured.  You  discuss  the  sensitivity  to  $\gamma_0$  in 1st paragraph and then discuss the  sensitivity to  different  forcings under a medium value of $\gamma_0$ for the  rest  of  the  section. However, about the sensitivity to different forcings, you jump between different regions, different GCM forcings, low and high emissions, which is quite chaotic. The focus of  each paragraph is not clear to me. About the summary paragraph in Line 245-246, it's a repeat of caption rather than a summary. I think Fig 9 is a good plot which can help you explain things earlier, which should be combined in your result description earlier.

Thank you for your comment. We will take everything into consideration in the revised manuscript version. Taking into account another comment from another referee, we will move the beginning of section 3.1 (results in the initialization proceeding) to a new section 2.5 at the end of the Methodology section.

We will add subsections for each part to make it more clear. Regarding Figure 9, we will refer to it before, when talking about sensitivity.

The Sec. 3.3 describes the individual effect of the atmosphere and the ocean on the projected ice mass loss from west and east Antarctica. However, it is not enough evidence to say that atmosphere is the dominant factor to margin of East Antarctic Ice Sheet, especially for Cook Ice shelf region, Toten Glacier region and Amery Ice Shelf region. These regions are losing ice clearly and show rounding line retreat in Fig 13b. Although it is not as obvious as west antarctica, it should be discussed at least.

In Figure 13, we wanted to make the argument that the atmosphere-only forced simulations (Fig. 13a) produce increases in ice thickness mainly in the AIS interior, while ice margins and shelves generally decrease.

Meanwhile, Figure 13b shows the disappearance of the ice shelves not only in the WAIS but also in the EAIS, with barely any increase in ice thickness. In this case, Pine Island, Ross and Ronne-Filchner retreated in a great amount towards the interior. For the atmosphere-only case, the ice-shelf retreat is much more limited. Looking at the numbers included in the figures for each experiment, we can see that the contribution of the ocean separately is between 2 and 3 metres for the highest emissions scenarios, mainly due to the WAIS retreat. Nevertheless, the total ice mass loss when both atmospheric and oceanic forcings are considered is not simply the sum of the two contributions, because nonlinear processes exist and play a role in the flow of ice.

In the revised manuscript, we will add this discussion and clarify in a better way the differences between these cases for both the EAIS and WAIS.

There are 13 figures and 4 tables in this manuscript, some of the figures are not very crucial to be included in the main manuscript, which can be moved to the supplementary material.

We agree with the reviewer. We have therefore decided to move Figures 8 and 11 to the supplementary material in the new version.

The resolution of some figures (Fig. 7, 8, 9, 11) is pretty low and needs to be improved. Some of the text labels on the figures appears. due to the low resolution of figures.

Yes, we have significantly improved the quality of the figures in the new version of the manuscript.

**Specific Comments:**

We will take special care to address each of the specific comments in the next version.

L21-22: citation please

Figure 1 in Rignot et al. (2013).

L28: What is 'ejecting circa'?

"ejecting circa 3331 Gt of ice to the Amundsen Sea". We meant that the Amundsen Sea has received approximately 3331 Gt of ice.

L45-46: need to add citation Edwards et al., (2021). We will include it in the next version.

L50: ice sheet stability? Instability could also be considered, but we referred to stability as a general concept, including both stability and instability.

L68: ISMIP6 is short for The Ice Sheet Model Intercomparison Project for CMIP6. We will clarify this in the next version.

L115: citation for this regularized Coulomb sliding law please. Schoof, C.: The effect of cavitation on glacier sliding, P. Roy. Soc. A-Math. Phy., 461, 609–627, https://doi.org/10.1098/rspa.2004.1350, 2005.

L130: you should explain specifically what $\gamma 0$ is here. $\gamma 0$ is the heat exchange velocity between ice and ocean, a parameter relating the rate at which the ocean and the ice in the shelves transfer their heat. We will clarify this in the updated version.

L163: is → are We will change it, thank you.

L169: which GCM forcing is chosen here for period 1995-2014? For the atmospheric fields, Surface Mass Balance (SMB) and 2m-air temperature, we have used monthly values from the Regional Atmospheric Climate Model (RACMO v2.3) together forced by ERA-Interim reanalysis mean climatology dataset from 1981-2010. Regarding the oceanic fields, the reference is Jourdain et al. (2020). We will clarify this in the final version.

L181: (Lipscomb et al., 2021) → Lipscomb et al., (2021) We will change it in the new version.

L190: imposed → kept constant? Yes, this is what we meant.

L198-199: refer to Fig. 4a here. Done.

L205: ocean forcing from CESM2 is not the maximum here compared with UKESM and HadGEM2. In this figure (Figure 1b), the grounding line is fixed, not evolving. Therefore we were referring to Figure 1c where we take the Yelmo outputs and the grounding line evolves.

L208: 'the low-emission case' with UKESM. No, it is rather the high-emission case. When referring to the low case we always add low at some point to distinguish.

L213: suggest 'we can see the sea level contributions slows down in the last two centuries.' Thank you for your suggestion. We will incorporate it in the revised manuscript.

L215: I think they are all nearly above 5 mm SLE/yr while HadGEM2 is close to 10 mm SLET/yr. Yes, we will change this to clarify (at 2500).

L219: for the medium value of $\gamma_0$, no experiments reach 3 m at 2500. Indeed, it was 2 m, not 3 m. The 3 metres were for the general case of the AIS. We will change that.

L221-222: 75% + 20% is not 100%. These were general numbers trying to summarize roughly for all the models, but only taking CESM2-WACCM and HadGEM2-ES in consideration. We will correct this in the new version.

L227: Fig 1 did not show anything related with Amundsen and Bellingshausen sea sectors. It indeed was Figure 2, we will correct this.

L228: for Amery, you need to cite Fig 7 rather than Fig 6. Indeed, we will change it.

L230: why do you only mention Ronne-Filchner here? Amery also shows loss of ice only for CSEM2 and CCSM4. Yes, we will correct this in the new manuscript. .

L233: it is not obvious to me in 2200. 2300 is more obvious. We will change the description in the revised text.

L235: not for CCSM4 Indeed, we will clarify that.

L236: what do you mean "reaching well above 300m"? the thickness? Yes, ice thickness. We will clarify this.

L237: why do you exclude the PIG and Thwaites here? We will include them.

L240: what do you mean "Ronne-Filchner ice shelf is larger"? If you are talking about readvance of GL (which I can't tell it well from current resolution of figure) in these regions, the ice shelf area is smaller.

No, we meant that for the low-emission case, the Ronne-Filchner ice-shelf is gaining ice (thickness) with respect to the start of the simulation. We will make that sentence more clear.

L242: The ice flow is decelerating in ROSS under low emission scenario. Yes, the ice is decelerating in Ross in that case. See next answer here.

L253: this confused me a lot. If the ice shelf disappears from 2300 onward, why we still see floating regions in Figure 7 after 2300. From Figure 7, the ice shelf is always there.

Yes, we admit this can  be a little bit confusing. In Fig. 7, we decided to plot the anomalies w.r.t the start of the forcing but without masking the ice as it evolves, that is, keeping the original size of the ice sheet. In this way, it is more apparent  where the

loss of ice takes place (mainly in the WAIS), identified by red colors. Masking the ice as it evolves instead is done on Fig. 8 for ice surface velocity.

L263: I can not tell mass loss from WAIS changes its tendency from Fig 11.

Yes, this was a mistake from our side, we meant Fig. 10.

L279: When you say 'The EAIS ice shelves also have their ice mass reduced', which plot are you talking about? Or you mean the margin of EAIS? We are talking about Figure 13. We meant both that ice shelves have a negative anomaly compared with 2015 and in some cases like CESM2-WACCM the margins also retreat towards the interior. We will clarify this issue in the new version.

L280: 'in the interior and eastern areas' → 'in the eastern interior'? We will change this.

L297: the forcing is stopped? I thought it was just kept constant. Yes, it is kept constant from 2300 to 2500. We will clarify this, "stop" is not the right word.

L312: in more than 3 meters → by more than 3 meters; 'affecting predominantly to the WAIS' →  'affecting the WAIS predominantly' Thank you for revising the language, we will change those sentences.

L314: I don't understand this sentence. Do you mean through enhanced surface melting?

No, we meant enhanced accumulation of ice. Figure 13a shows that the eastern region is gaining ice and therefore it is an opposed phenomenon compared with the ocean effects. We will make this more clear.

L319: There is also a clear mismatch between your result and Greve et al. 2022 in Table 4. Yes, we will mention that.

L321: Then what about the mismatch in CCSM4?

CCSM4 shows a mid-range air temperature evolution compared  to the other GCM forcings, but it shows much lower ocean temperatures and therefore thermal forcing (Figure 1). The average anomaly in thermal forcing on the shelves in 1995-2300 with respect to the reference climatology for Greve et al. (2022) (see Table 1 in their article) is 1.652 ºC for both CCSM4 and 1.613 ºC for CESM2(-CAM), indicating very similar oceanic forcings for these two models in the  extension towards 2300.  Their results for sea level contribution are accordingly very similar for these models. In our case, these numbers are 1.586 ºC for CCSM4 and 2.373 ºC for CESM2-WACCM, calculated as the average thermal forcing under the shelves with respect to the reference given by Jourdain et al. (2020). Therefore, the ISMIP6-2300 extension of CCSM4 has a smaller thermal forcing than the extension made by Greve et al. (2022).

 L346: unfished sentence. It is unclear to us what the reviewer meant here.

L366: on Eq (3). Or you don't need to mention it here. We will revise that for the next version of the manuscript.

L372: I think you mean a position contribution to SLR. But by saying 'negative effect' can be confusing. Yes, we meant that. We will rephrase it for the next version.

**Figures**

Figures have been redone trying to make them larger while keeping the resolution and some extra features have been added following the advice of both referees. In the rewritten text these new figures will appear and we have thought about including some of them as an Appendix or Supplementary Material.

Figure 2: Caption: the differences in thermal forcing for b). We will change it, thank you.

Figure 7 & 8: I did not see the coastlines at all, which is a very important feature to be shown clearly in the figure. Yes, we have changed it. Nonetheless, we consider that only representing the grounding line improves the visualisation of the plot.

Figure 8: what do you mean the cells where there is ice? I think you mean grounded ice? If yes, why the floating ice is included in UKESM LOW emission case?

No, we mean both grounded and floating ice. For higher emissions experiments, the floating ice starts to disappear in 2200. That is why some ice shelves turn white (ocean has made its path here) but in the low-emission scenario there is still ice on the shelves. We only wanted to show the retreat towards the interior and the ice accelerates near the grounding lines (and of course where there is ice).

[Figure]

Figure 7. Ice thickness anomalies (in m) for years 2100, 2200, 2300, 2400 and 2500 with respect to the start of the simulations for the set of five experiments carried out with the median value of $\gamma_0$. The grounding line is represented by black and violet colors for 2500 and 2015 respectively. To show negative anomalies in the WAIS, no masking has been implemented as time evolves, keeping the original coastlines.

[Figure]

Figure 8. As Figure 7 but for ice surface velocity anomalies (in m/yr). Note that only the cells where there is ice (floating and grounded) at the specific times are represented.

Figure 12: in the AIS → from the AIS We will change it.

**References**

Thank you, we will include your given reference in the new version and refer to it at the lines you have said. Note that there are some others (in blue) cited by us in our reply.

Berdahl, M., Leguy, G., Lipscomb, W. H., Urban, N. M., & Hoffman, M. J. (2023). Exploring ice sheet model sensitivity to ocean thermal forcing and basal sliding using the Community Ice Sheet Model (CISM). *The Cryosphere*, *17*(4), 1513-1543.

Blasco, J., Tabone, I., Alvarez-Solas, J., Robinson, A., and Montoya, M.: The Antarctic Ice Sheet response to glacial millennial-scale variability, Clim. Past, 15, 121–133, https://doi.org/10.5194/cp-15-121-2019, 2019.

Blasco, J., Alvarez-Solas, J., Robinson, A., and Montoya, M.: Exploring the impact of atmospheric forcing and basal drag on the Antarctic Ice Sheet under Last Glacial Maximum conditions, The Cryosphere, 15, 215–231, https://doi.org/10.5194/tc-15-215-2021, 2021.

Edwards, T.L., Nowicki, S., Marzeion, B. et al. Projected land ice contributions to twenty-first-century sea level rise. Nature 593, 74–82 (2021). htps://doi.org/10.1038/s41586-021-03302-y

Lipscomb, W. H., Price, S. F., Hoffman, M. J., Leguy, G. R., Bennett, A. R., Bradley, S. L., ... & Worley, P. H. (2019). Description and evaluation of the community ice sheet model (CISM) v2. 1. *Geoscientific Model Development*, *12*(1), 387-424.

Lipscomb, W. H., Leguy, G. R., Jourdain, N. C., Asay-Davis, X., Seroussi, H., and Nowicki, S.: ISMIP6-based projections of ocean-forced Antarctic Ice Sheet evolution using the Community Ice Sheet Model, The Cryosphere, 15, 633–661, 2021.

E. Rignot *et al.*, Ice-Shelf Melting Around Antarctica.*Science* 341, 266-270 (2013). DOI:10.1126/science.1235798

Schoof, C.: The effect of cavitation on glacier sliding, P. Roy. Soc. A-Math. Phy., 461, 609–627, https://doi.org/10.1098/rspa.2004.1350, 2005.

Sutter, J., Jones, A., Frölicher, T. L., Wirths, C., & Stocker, T. F. (2023). Climate intervention on a high-emissions pathway could delay but not prevent West Antarctic Ice Sheet demise. Nature Climate Change, 13(9), 951-960.

---

## Author Response (AR2)

**Replies to the 2nd round of comments from referees 1 and 2 of: "Antarctic sensitivity to oceanic melting parameterizations"**

Antonio Juarez-Martinez, Javier Blasco, Alexander Robinson, Marisa Montoya and Jorge Alvarez-Solas

This document contains the responses to the comments made by Referee 1 firstly and by Referee 2 secondly, for their second report in regard with the revised version of our manuscript. Note that our comments are in blue.

In the text below, we respond to the specific points raised by the referees. Apart from the comments, we have changed some citations that have changed since our last submission, especially the reference to the ISMIP6-2300 dataset.

Referee 1 comments:

I would like to thank the authors for their careful considerations from the first round of reviews they have received. I believe they addressed most of my comments. The extensive new text added to the manuscript is leading me to make additional minor comments.

We would like to thank the reviewer for helpful and thorough comments which have helped to improve our manuscript. In addition, we know now that Seroussi et al. (2024) has been accepted, so following your suggestion in your previous report and after personal communication with Dr. Seroussi and the ISMIP6 team, we will change the reference for the new ISMIP6-2300 dataset.

I will begin by making a style comment that is personal (I hinted at it in a few places in the first round of review). Many parts of the manuscript are written using a passive voice and I find it, at times, unnecessary and making a sentence more difficult to read than it should. I would encourage the authors in the future to try to use a more active style which (in my opinion) helps with the flow and the delivery of the information. Again, this is a personal taste.

We have taken into consideration using a direct voice when possible.

Below you can find the one-by-one response to your technical comments.

**Technical comments**

**Line 134: remove the first reference to "(Lipscomb et al. 2019)". The one at the end of the sentence is enough. Also, add the limit in ice thickness you are using similarly to your response in my first review.**

We have deleted the first reference and added the limit for ice thickness (200 m) as in the first review.

Line 136: define the acronym for ELRA.

We have defined ELRA as the Elastic Lithosphere Relaxing Asthenosphere method.

Line 144: replace the citation of (Seroussi et al., 2020) by (Favier et al., 2019).

Done.

Line 150: rewrite "where it has been denoted by TF(x,y,zdraft)" by "where TF(x,y,zdraft) denotes". Also, z_draft is not the thickness of the ice shelf but rather the depth of the ice shelf base (or the draft of the ice shelf).

We have redefined that sentence in the new version and clarified the definition of z_draft conveniently. Thank you for this correction.

Line 168: please, define the acronym AOGCM.

We have defined it as Atmosphere-Ocean General Circulation Model in the new manuscript version.

Line 177: "Therefore,…" I will reiterate my statement from my first round of review. The ocean thermal forcing does not react independently from the atmospheric condition (given that is directly linked to ocean temperature and salinity). The way I interpret your sentence is such that if you were to warm atmospheric temperature by say 4 degrees worldwide (using various mechanisms) in CCSM4, the ocean will not react to this change. I find it very hard to believe.
If, for similar forcing, a model yields a different ocean response, it could be tied to many aspects of the model including model resolution, model physics, model sensitivity, … Again, I would simply remove this statement which does not add to any points you are trying to make in this paper.

We agree with this suggestion; therefore we have removed it in the current version.

Line 177-180: this sentence is a direct repetition of the previous one. I would remove it.

Yes, we have removed it.

Line 182-184: this seems to be out of context here because the evolution of grounding lines is dependent of your model and its transient behavior. Are you trying to say that in WAIS, many places of the bed topography are reversed sloping, and the grounding line will retreat to location with increasing thermal forcing?

Yes, indeed, we wanted to illustrate that when the ice sheet retreats, the grounding line evolves and thus the thermal forcing changes. We will add that information to the sentence.

Line 185: Looking at Figure 2a, it looks to me that the temperatures in the Weddell sea and over the Ronne-Filchner ice shelf have risen more compared to the regions you

are referencing. I would simply rephrase as: "The temperatures over Ronne-Filchner and Ross ice shelves and the Amundsen Sea rose more than 15K …"

You are right, we will change that sentence for the new version.

Line 206: replace "variables" by "ice thickness and bed elevation"

Yes, it is more specific as you say. We will change that for the next version.

Line 226: The content here does not deserve its own section. I would simply move it to section 2.3 where you describe your experimental protocol. Otherwise, move your entire experimental protocol here.

Yes, after the comments of both referees the section has been reduced to three lines so we have decided to move it to section 2.3 in the next version of the document.

Line 229: replace "held until 2500" by "held constant until 2500".

We have replaced it as it is important to clarify that.

Line 240: the ice shelf velocities are overestimated almost over all ice shelves except Ronne-Filchner. Please add this distinction.

Yes, indeed. We have made the distinction to give that information.

Line 244: replace "climate" by combined "atmospheric and oceanic".

We have decided to rename this section titled as: "Sensitivity to combined atmospheric-oceanic forcing and $\gamma_0$ value"

Line 267: rewrite "On a regional basis," as "Regionally,".

We have changed  for the next version of the manuscript.

Line 268: I believe you mean "more than 2m" as opposed to "up to 2m".

Yes, it is what we meant. We have changed it.

Line 275: replace "the reasons behind the behaviour of the time series shown" with "our results".

We have done that for the new version.

Line 277: replace "(Figs. 8 and A4, respectively)" with (Figs. 8, A4, A5).

We have replaced that reference to the figures as suggested.

**Line 279-280: simplify the sentences by removing some of their text and writing something like: "the EAIS also loses ice particularly in the margins of the ice sheet, while…"**

We have changed that sentence by: "the EAIS also loses ice mass particularly in the margins of the ice sheet (e.g. the Amery ice shelf), while there is gain of ice in the interior. "

**Line 284: replace "between 2200 and 2300" by "prior to 2100". In the figure you can see the groundling has moved already at 2100.**

Yes, but it becomes more clear from 2100. Nevertheless, you are right and we have changed the sentence.

**Line 293: replace "in the sense that is gaining ice" with "in the sense that it is gaining ice".**

Thank you for the revision, we have added "it".

**Lines 344-350: this is a direct repetition of your results. I would suggest removing this paragraph since your discussion is now a bit long. I don't have a strong feeling about this.**

We have removed the entire paragraph.

**Line 374: "that could explain this difference" This would explain it if the resulted forcing were different. Please rephrase.**

We think that the difference lies in the warm biases in CESM2-WACCM (Purich and England, 2021) due to the representation of the cloud feedback (Zhu et al., 2021). We have addressed this in the Discussion section, but in this particular sentence (Line 374) we will make reference to this as these two articles point to the fact that CESM2-WACCM is a model producing more warming than others.

**Line 384: the underestimation of Lipscomb et al. (2021) is only true if you consider your model to be the truth. Maybe you meant to say their results is underestimated compared to yours? Please rephrase.**

We wanted to point out that Lipscomb et al. (2021) use data from the 21st century that in a changing climate could not be entirely realistic and therefore could be underestimating the future emissions that the climatic component will experience. Nevertheless, we will follow your suggestion and rephrase the sentence in order to incorporate the comparison to our results when referring to the underestimation issue. There is also a comment on the Introduction that has changed after this suggestion.

**Line 384: remove the first "but".**

Thank you for noticing, there were two "but" and we have removed the first one.

**Line 407: replace "contrary" with "opposite".**

Thank you for the language revision here. We have changed it.

**Line 408: replace "with 16" with "using 16" and remove the word "grids".**

Yes, we have changed that and removed the word "grids" in that sentence.

**Line 409: replace "nonetheless" with "while".**

We have replaced it and it is more clear now. Thank you for the correction.

**Line 410: remove the word "observed".**

We have deleted that word.

**Line 410: "This suggests…" Mentioning MISI suddenly is a bit far-fetched without finer analysis. I would remove this sentence.**

OK. We have removed that sentence. We took that sentence from Sutter et al. (2023), maybe just referring to it is enough to clarify.

**Line 428: replace "and that results" with "resulting"**

We will do it for the next version.

**Tables**

**Table1: you are running more than 5 experiments in your work. I would modify this table to capture them all. As is, this table mentions the forcing and model used for the experiment. You could add 3 columns for each set of experiments: 1 column for "experiment forced with atmosphere + ocean", 1 column for "experiments with ocean only" and another one for "forcing with atmosphere only".**

Here we think that there is a linguistic problem. As an experiment we mean the model used, that it is certain that has been used three times (atmosphere-ocean combined, ocean-only and atmosphere-only). If we modify the table as you say, we will see the same information in all columns as the Scenario and Model does not change, only the activation or deactivation of the oceanic/atmospheric component which can be said by words as in a previous section (2.4 in the revised manuscript). We have clarified in a better way in the table caption saying: "Description of the five types of experiments carried out based on the protocol for ISMIP6-2300…." and in the paragraph mentioning the "only" cases subset of experiments. See also our next comment for Figure 1.

**Figures**

**Figure 1: these forcings are also used in your other 10 experiments, right? If so, correct the "five experiments" in the caption. Also, in the caption, I would suggest rewriting "evolving grounding lines in the AIS (c)" by "evolving grounding lines (c) in the AIS". Your version made me believe that the word "AIS" was only for panel (c).**

As in the last technical comment, we have only considered five experiments (those in the legend and in the caption) corresponding to the five different models, but it is true that we have used them three times for each type of simulation. We have clarified in a better way in the caption saying: "the five models considered for simulation: …
We have changed the (c) to clarify as you have suggested.

**Figure 7: in the caption, rewrite "since the start…" with "between 2015 and 2500".**

We have taken into account this suggestion in the new version.

**Figure 8: in the caption, replace "for the years…" with "for years…"**

Thank you for your correction. We have done that in the new version.

**References**

**Favier, L., Jourdain, N. C., Jenkins, A., Merino, N., Durand, G., Gagliardini, O., Gillet-Chaulet, F., and Mathiot, P.: Assessment of sub-shelf melting parameterisations using the ocean–ice-sheet coupled model NEMO (v3. 6)–Elmer/Ice (v8. 3), Geoscientific Model Development, 12, 2255–2283, 2019.**

Referee 2 comments:
**Juarez-Martinez et al. has addressed most of the comments and suggestions well. The modified version looks much better. However, there are another three comments which need the authors' attention:**

Thank you for your comments and for helping to make this research article more structured and appropriate. Please, find the one-by-one reply to your comments and suggestions in the next lines.

**1. Numbering of subtitles in Sect 3.1 and Sect 3.2: Why do you need a sub-subtitle if there is only one sub-subsection? For example, if you only have 3.1.1, why not just keep it under 3.1? Same for 3.2.1.**

We thought that separating them was more suitable to differentiate the several contents based on your previous report comments. In the case of 3.1.1, it allows to separate the general AIS results from the WAIS and EAIS, for example. Same happens for 3.2.1, as it focuses on the WAIS, and in our criterion makes it more clear to the reader to separate. Nevertheless, we have removed the numbering of sub-subsections while keeping their title just to separate the parts as we have commented before.

**2. Putting the numbering aside, it's not clear about the difference in focus if you look at the subtitles of Sec 3.1 and Sec 3.2. I think Sec 3.2 is focused on spatial sensitivity to different climate forcing, which is a better subtitle than "Spatial patterns in the loss of ice" and comparable to subtitle of Sec 3.1.**

We have taken that into account and renamed the above-mentioned title of the section.

**3. In the discussion about the difference to Lipscomb et al 2021, you used different ways to extend the forcing between 2300-2500. You mentioned Lipscomb et al. (2021) considered the climate variability, which is more reasonable compared to yours. In such case, you should say your result overestimates the AIS response rather than Lipscomb et al. (2021) underestimates the AIS response.**

With the underestimation of Lipscomb et al. (2021) we were referring to the fact that they use only the last years of the 21st century to make the extension which could underestimate future emissions that we can capture with the data from 2100-2300.
Anyhow, we have changed the text in accordance with another referee to make clear that with "underestimation" we intend to compare with our results and therefore avoid a misunderstanding.